# Choice of method of place cell classification determines the population of cells identified

**Dori M. Grijseels**[1]*, **Kira Shaw**[1], **Caswell Barry**[2], **Catherine N. Hall**[1]*

**1** School of Psychology and Sussex Neuroscience, University of Sussex, Brighton, United Kingdom,
**2** Research Department of Cell and Developmental Biology, University College London, London, United Kingdom

* D.Grijseels@sussex.ac.uk (DMG); catherine.hall@sussex.ac.uk (CNH)

**Data Availability Statement:** All data files are available from the Figshare repository (DOI: 10.6084/m9.figshare.13560548). Model generation and analysis code is available from the GitHub repository (https://github.com/DoriMG/place_cell_

## Abstract

Place cells, spatially responsive hippocampal cells, provide the neural substrate supporting navigation and spatial memory. Historically most studies of these neurons have used electrophysiological recordings from implanted electrodes but optical methods, measuring intracellular calcium, are becoming increasingly common. Several methods have been proposed as a means to identify place cells based on their calcium activity but there is no common standard and it is unclear how reliable different approaches are. Here we tested four methods that have previously been applied to two-photon hippocampal imaging or electrophysiological data, using both model datasets and real imaging data. These methods use different parameters to identify place cells, including the peak activity in the place field, compared to other locations (the Peak method); the stability of cells' activity over repeated traversals of an environment (Stability method); a combination of these parameters with the size of the place field (Combination method); and the spatial information held by the cells (Information method). The methods performed differently from each other on both model and real data. In real datasets, vastly different numbers of place cells were identified using the four methods, with little overlap between the populations identified as place cells. Therefore, choice of place cell detection method dramatically affects the number and properties of identified cells. Ultimately, we recommend the Peak method be used in future studies to identify place cell populations, as this method is robust to moderate variations in place field within a session, and makes no inherent assumptions about the spatial information in place fields, unless there is an explicit theoretical reason for detecting cells with more narrowly defined properties.

## Author summary

Place cells are hippocampal cells that have spatially constrained receptive fields, the place field. These cells have been widely studied in the context of navigation, more recently using virtual reality environments in combination with optical methods of recording neuronal activity. However, there is a lack of consensus regarding how to identify place cells in these data. In this study we tested the sensitivity and specificity of four methods of

methods), or linked from www.brainenergylab.
com.

**Funding:** This work was supported by a Sussex
Neuroscience PhD studentship for D.M.G, a MRC
Discovery Award (MR/S026495/1; mrc.ukri.org)
for K.S. and C.N.H, an Academy of Medical
Sciences/Wellcome Trust Springboard Award
(acmedsci.ac.uk) for C.N.H., a MRC Project grant
(MR/S026495/1) for C.N.H and a Wellcome Senior
Research Fellowship (212281/Z/18/Z; Wellcome.
org) for C.B.. The funders had no role in study
design, data collection and analysis, decision to
publish, or preparation of the manuscript.

**Competing interests:** The authors have declared
that no competing interests exist.

identifying place cells. By comparing these methods and quantifying the populations of place cells they identify, we aimed to increase our understanding of exactly the populations that are currently being studied under the name "place cells". Although the appropriate method may depend on the experimental design, we generally recommend a single method going forward, which will increase consensus within the field about what should be included in a place cell population, and allow us to better compare results between studies.

## Introduction

Place cells are a subset of hippocampal pyramidal cells that fire selectively when the subject is in a certain location [1], and provide a sparse population code for self-location. Studies revealing their properties, including location-specific firing [1], directional selectivity [2] and context-dependence [3] have been vital for our understanding of how the hippocampus codes space. [4]. Place cells are characterised by their place field, which is a spatially stable location where the cell preferentially fires. Depending on the size of the environment, place cells can have one or multiple place fields [5]. The place fields may change location between different environments, a phenomenon called remapping. Place cells are also relevant to understanding clinical conditions such as Alzheimer's disease, as in mouse models of Alzheimer's disease they show impaired firing [6] linked to memory deficits.

Since their discovery, place cells have been extensively studied in real world environments–both open field and constrained—using electrophysiological recordings while animals either explore freely or perform directed spatial tasks. In these studies, cells are generally included in analyses based on a variety of properties, including their peak firing rate, waveform, sparseness of the place field or spatial selectivity (e.g. [3,7–9]). Further analyses sometimes require the place fields of these cells to have additional properties, such as a maximum number of place fields allowed, or a stable place field over time (see e.g. [7,8,10]).

However, methodological advances now also allow place cells to be studied in vivo using calcium imaging [11–13], enabling large populations (n>100) of cells to be recorded for multiple sessions. This method requires the brain to be stationary during recording, which necessitates the use of a Virtual Reality (VR) environment. Often a visual VR is used during imaging, consisting of one or multiple screens displaying an environment that the mouse can control by running on a ball or wheel. With a ball, the mouse is able to move in two dimensions (e.g. [14]), whereas with a wheel, mice can only move backwards and forwards. These types of VR limit the animal's ability to look around, and provide sparse sensory feedback; for example, whisking does not provide information about the environment. Place cells also respond differently in VR environments compared to the real world, showing broader place fields and increased directionality [15].

Several studies using one-dimensional environments (i.e. corridors) have revealed that hippocampal pyramidal cells represent other features in addition to location, such as reward [16] and travelled distance [17]. It is unclear to what extent these features are coded by separate populations of cells in the hippocampus or cells that, under other circumstances such as a dedicated navigation task, might be recruited as place cells. To resolve this, it is important to be able to accurately define place cells within such one-dimensional environments, but currently different studies use widely varying methods making comparisons between studies problematic.

Varying definitions of place cells have been chosen to account for the constraints imposed by the imaging methodology. Unlike electrophysiological recordings, imaging detects changes in intracellular calcium levels rather than direct readouts of action or synaptic potentials. The non-linear relationship between calcium transients and spike rates makes it hard to accurately estimate spike rates from two-photon data [18], so adopting exactly the same methods as used in electrophysiological recordings is not possible. In addition, the exact waveform of the spikes, another characteristic used to classify place cells (e.g. [7]), is unknown. Instead, studies have tended to use adaptations of some, but not all, of the imaging equivalents of peak firing rate, sparseness of off-location firing, and place field stability to define a cell as a place cell: Dombeck et al. [11] categorised a cell as a place cell based on a combination of properties of that cell's apparent place fields, including the size, peak calcium fluorescence and the ratio of the firing within and outside the field. This method, or variations of it, have subsequently been adopted in several studies (e.g. [19–21]). Fournier et al. [22] proposed a statistical method that uses the peak activity in the rate map of a cell and compares it to the peak activity in shuffled versions of the cell. An alternative approach [23] detects place cells from the stability of their activity as a function of location, a method based on those used in electrophysiology experiments [3]. Lastly, Ziv et al. [24] defined place cells based on the mutual information between the cells and the location of the animal, a method which is also used widely [25–27]. However, it is unclear what biases these different methods exhibit and to what extent their classification criteria are equivalent–in short, do they identify the same neurons as place cells?

In this paper we aim to address the lack of consensus on how to identify place cells in two-photon data in a one-dimensional VR environment. We compare the performance of two established methods for the identification of place cells in two-photon data (the methods used by Dombeck et al. [11] and Ziv et al. [24]), another method that is often using to characterize place cells in electrophysiological studies (e.g. [3,7], described by O'Leary et al. [23]) as well as one method developed for use on electrophysiology data which has been adapted for use in imaging–described by Fournier et al. [22]. We applied them to a range of synthetic model cell populations to explore whether cells identified as place cells by each of these methods possess similar characteristics. We conclude that in a range of mock datasets the method developed by Fournier et al., which we call the Peak method, is best suited to identify place cells, having a high sensitivity and specificity, and lacking assumptions about spatial information held by the place cells. As a result, we recommend this method for the identification of place cells in two-photon imaging data. Our data also show that choice of identification method is important, as the methods classify different, largely non-overlapping populations of cells as place cells.

## Results

### Place cell detection by the Combination and Stability methods is sensitive to the properties of place fields and the number of times mice traverse the environment

We evaluated the performance and suitability of four different approaches for detecting place cells (S1 Fig):

1. The Peak method, which classifies cells as place cells based on the average rate of firing (approximated by fluorescence change in two-photon data) in one location being higher than in the rest of the environment [28];

2. The Stability method, which identifies place cells as those with stable firing patterns across locations over time [23];

3. The Combination method, which requires place cells to fire more across a contiguous stretch of the environment than at baseline and to fire in this location on at least 20% of traversals [11].

4. The Information method, which classifies cells based on the increased amount of spatial information the cells hold about the animal's location [24].

We selected the four methods for varying reasons. The Peak method was originally applied to electrophysiological data, and its simplicity allows for the application to fluorescence data with minimal adaptations. Similarly, the Stability method was based on methods used for characterising place cells in electrophysiological recordings (e.g. [3]), while the Information method, or variations thereof, have previously been used to detect place cells in both electrophysiological [26] and fluorescence data [27]. This makes these three methods promising candidates for application across the two recording paradigms. In contrast, the Combination method was designed specifically for fluorescence data, and has been widely employed in the field (e.g. [11,20,25]).

To characterise the four different approaches for detecting place cells we applied them to synthetic datasets consisting of 20 true place cells and 80 random cells, matching the prevalence of place cells found using patch-clamp recordings [29]. Mouse location for each dataset was generated using experimentally acquired locomotion time courses from mice running through a linear virtual reality environment (S2 Fig, 8 datasets: 4 mice, 2 sessions per mouse). The dataset contained a total of 184 traversals of the environment, with mice running 23 traversals per session on average. Model fluorescence maps were generated by applying the synthetic cell profiles to 50 randomly selected traversals (Fig 1A and 1B). Periods where the mouse was not running–defined as having a velocity below 2 cm/s–were removed (Fig 1C and 1D).

Fluorescence maps of the place cells were modelled using a Gaussian to simulate changes in fluorescence as a function of location, where the mean determined the place field centre and sigma the field width. We used a default width of 50cm, based on widths previously reported in a virtual environments [11,29]. Synthetic data was linearly scaled to match the peak values observed in our own two-photon CA1 data (Fig 1E). We distributed place field centres evenly across the 200 cm long track, then convolved the place field with the position of the mouse to determine the fluorescence map of each cell over time (Fig 1E, bottom 20 cells, Fig 1F). We also included 80 control cells (Fig 1E, top 80 cells, Fig 1G) without place-dependence. Poisson noise, which in the default model had an average fluorescence of 0.18% of the default peak height, was then added to the traces of all cells. Lastly, the four different place cell classification methods were applied to these fluorescence maps.

We could therefore vary the parameters that determine the fluorescence map (Gaussian peak, width and location, number of traversals of the environment), each with a different effect on the properties of the model cells (S3 Fig), to determine how much the properties of the place field affect their detection by the different methods. To assess place cell classification performance, we first calculated how many place and non-place cells were correctly identified by each method (true positives and true negatives), versus the number of false positive and false negative identifications. From these values, we then calculated the sensitivity (the proportion of place cells that were correctly detected), and the specificity (the proportion of cells identified as place cells that actually were place cells) of each method (Methods, Eqs (1) and (2)). A perfect detection method would therefore have a selectivity of 1 and a specificity of 1.

## Number of traversal

First, we determined the impact of the number of times the mouse ran through the environment on place cell detection by varying the number of traversals used to generate the model

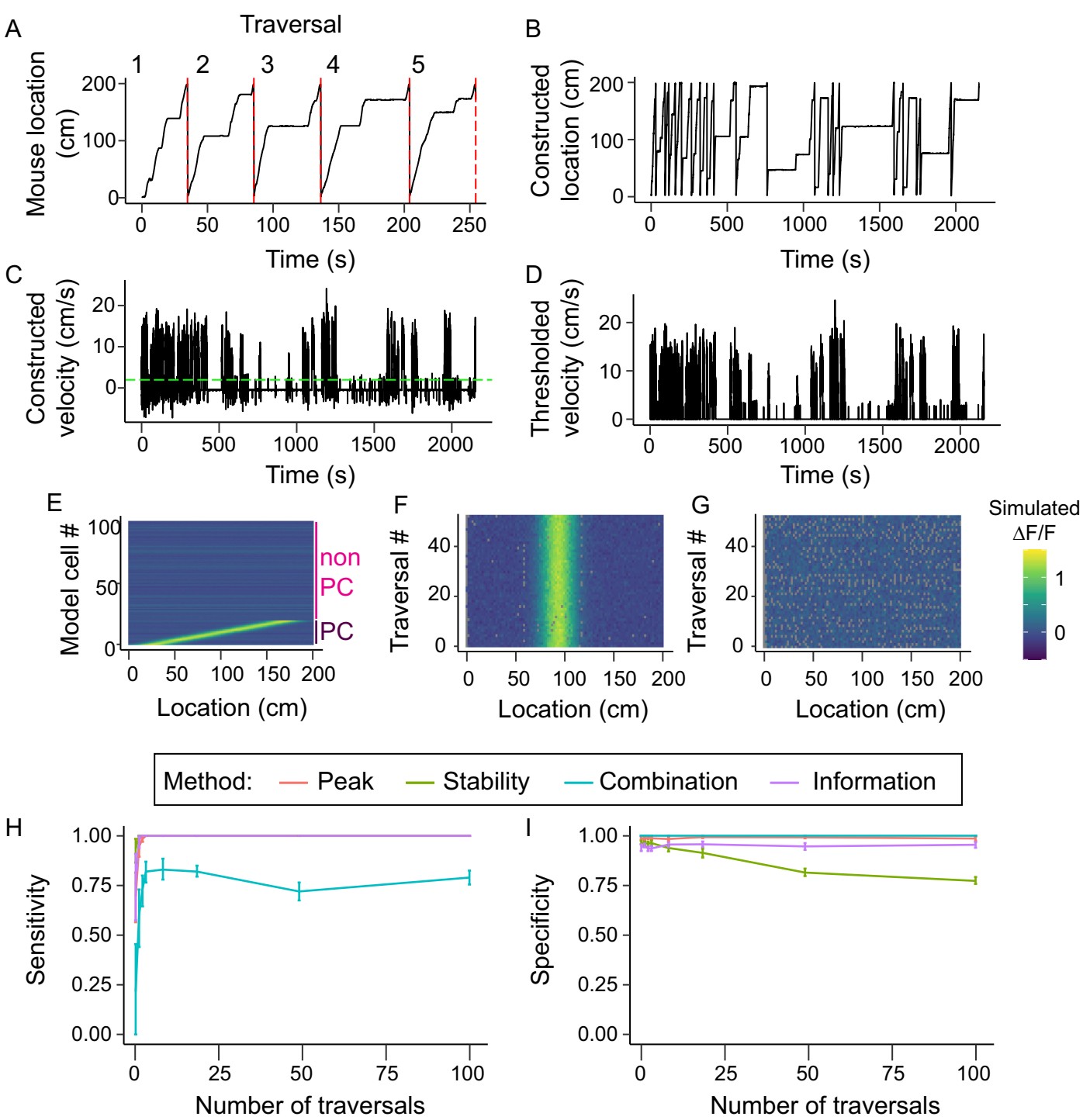

**Fig 1. Model locomotion profile generation and effect on methods.** (A) Section of a locomotion profile collected in a 200cm long virtual environment. This was cut into individual traversals of the environment (numbered, separated by red dashed lines) (B) locomotion profile generated from 20 randomly selected traversals. (C) Velocity profile of the locomotion profile shown in B. The green dashed line indicates the threshold for detecting running (2 cm/s) (D) Trace in C after thresholding. Time points with below threshold velocity were excluded from further analysis. (E) Fluorescence maps of 20 model place cells (cells 1–20) and 80 non-place cells (cells 81–100). (F) Activity over location of a single model place cell over 50 traversals. (G) Activity over location of a single non-place cell over 50 traversals. Grey points in F and G are missing data caused by the mouse across this location between acquisition of two frames. (H) The sensitivity of each method as a function of the number of traversals included in the locomotion trace. (I) The specificity of each method as a function of number of traversals. Lines in (H) and (I) show means over 10 randomly generated data sets. Error bars represent 95% confidence intervals.

datasets from 2 to 100 (Fig 1H and 1I). We generated 10 model datasets for each number of traversals.

The Peak, Stability and Information methods had a high sensitivity regardless of the number of traversals included in the dataset (Fig 1H). Both the Peak and Information methods also displayed a stable specificity across the range of traversals, with a mean of 0.99 and 0.95 respectively, reflecting their requirement that a true place cell's activity is in the top 1% and 5% (respectively) of shuffled data (Fig 1I). However, the Stability method saw a decreasing specificity with an increase in traversals, with a specificity of 0.76 at 100 traversals, i.e. more non-place cells were classified as place cells as the number of traversals increased. This is likely because with an increase in traversals, the Poisson noise of the non-place cells will increasingly average out over the traversals, thus resulting in an increased correlation between the fluorescence maps of the first and second halves of the session for the non-place cells. This increases the number of false positives, and thus results in a decrease in specificity. The Combination method increased in sensitivity as the number of traversals increased from 2 to 20, above which it stabilised with a sensitivity of 0.79. The specificity of this method remained 1 regardless of the number of traversals included.

Thus our simulations predict that the Combination method would fail to detect at least 27% of hippocampal place cells, while up to 23% of the place cells identified by the Stability method would be false positives, and accuracy for both methods is affected by the number of times mice run through the environment.

## Place field properties

We next tested the effect of manipulating the width of the model place fields and their peak "fluorescence" on the ability of the four methods to detect the place cells (Fig 2C–2F). First, we varied the width of the place field between 20 and 200 cm, equivalent to 10 to 100% of the total environment length, while keeping the peak value at a ΔF/F of 1.3 (see S4 Fig for example place cells). For the Peak, Stability and Information methods, varying place cell width did not materially affect sensitivity. However, the sensitivity of the Combination method was generally lower than the other methods, only approaching their sensitivity for place fields between 80 and 120 cm, and failing to detect any place field narrower than 40 cm or broader than 160 cm. Specificity of place cell detection was high for the Peak, Combination and Information methods and unaffected by the place field width, whereas it was generally lower when using the Stability method (Fig 2D). It increased linearly with place field width for this method, presumably because the correlation between the fluorescence maps of the non-place cells and place cells decreases as the width increases.

We then varied the peak of the place field between a ΔF/F of 0.0001 and 2 (equivalent to 0.008% to 155% of the average fluorescence map peak measured in our two-photon data), keeping the place field width at 50 cm. Sensitivity was only affected using the Peak, Stability and Information methods at extremely low, and therefore unrealistic, peak values ($< 0.1$, or fewer than 8% of peaks in our real data; Fig 2E). The sensitivity for the Combination method was generally lower and increased with place cell peak size. The specificity was not affected by changes in peak values for any of the methods (Fig 2F).

Next, we varied the number of place fields for each place cell, up to 4 place fields per cell (S5 Fig). The Peak, Stability and Information methods were able to identify the model place cells, regardless of the number of place fields, with a high sensitivity, while the Combination method showed a dramatic drop in sensitivity above 2 place fields. Specificity was relatively unaffected in all methods by increasing the number of place fields, though was maximal using the Stability method when cells had 4 place fields.

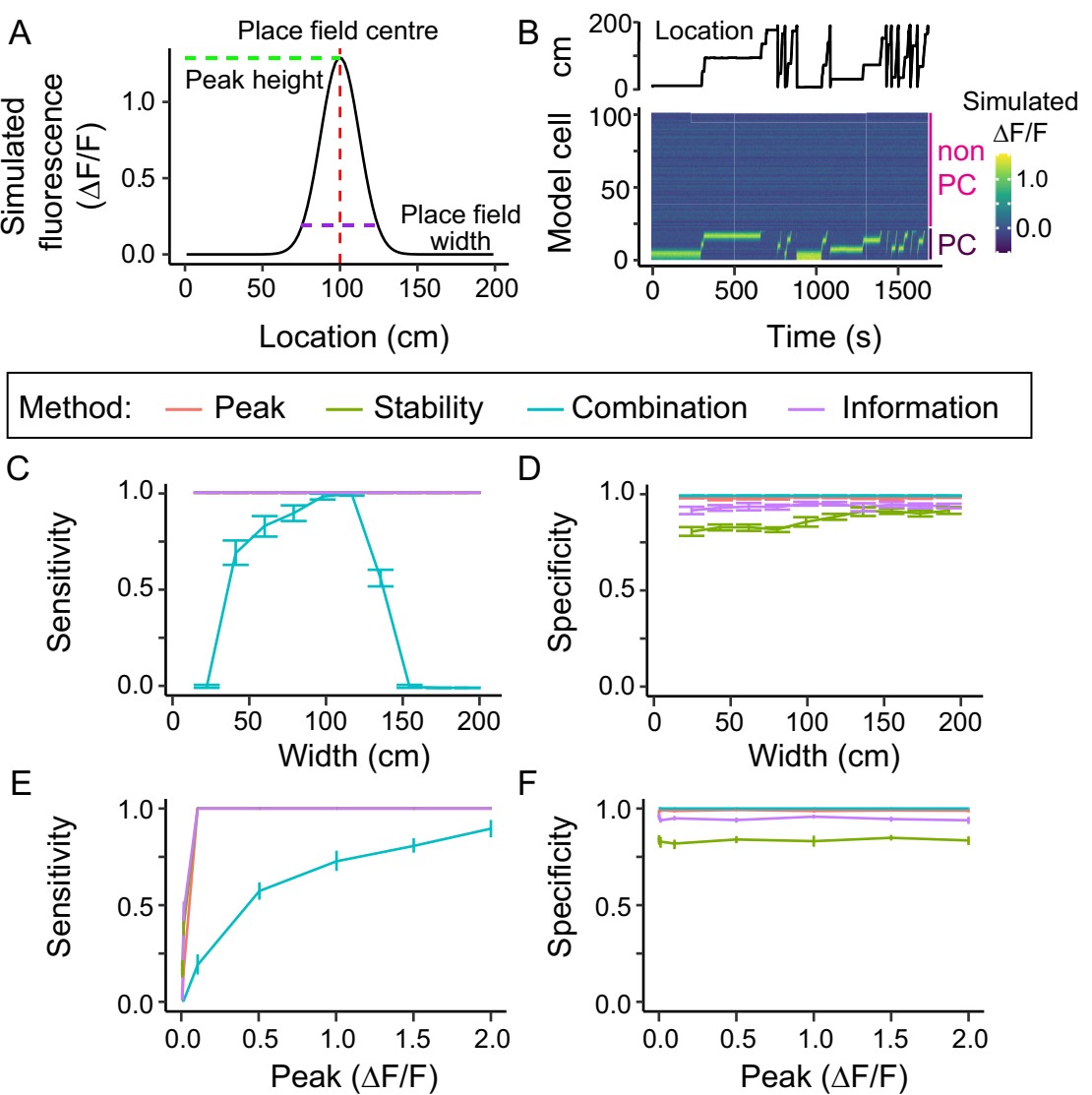

**Fig 2. Place cell model and its effect on place cell detection.** (A) Gaussian model of the fluorescence of a place field depending on the location (x-axis). The model takes the following parameters: the centre of the place field (red dashed line), the place field width (purple dashed line) and the peak of the Gaussian (green dashed line). (B) example fluorescence profile over time of a model data set containing 20 place cells (cells 1–20) and 80 non-place cells (cells 21–100), generated using the location trace shown. (C, E) Specificity and (D, F) sensitivity of the different methods as a function of the place field width (C, D) or peak (E, F). Note in D all methods except the Combination method had a sensitivity of 1 across all widths, and the lines overlap. In E there is similar overlap for peak values above 0.1. Data are means of 10 simulations. Error bars represent 95% confidence intervals.

Finally, we varied both the place field width and peak values at the same time (S6 Fig). The results reflect those when varying each parameter individually, illustrating the high general sensitivity of the Peak, Stability and Information methods and the much narrower performance of the Combination method, which has high sensitivity at only a narrow range of widths and with high peak fluorescence values. Conversely specificity is high overall for the Peak, Combination, and Information methods, but for the Stability method is lower across the range of parameters tested and lowest for narrow place fields.

In summary, the Combination method detected fewer model place cells and was the most affected by realistic variations in place field properties, while the Stability method classified

more non-place cells as place cells and this tendency increased with narrower place fields. Both the Peak and Information methods were sensitive and specific over the whole range of place field properties tested, though the Peak method slightly outperformed the Information method, having a higher specificity.

## Variability and reliability decrease detection of model place cells, particularly by the Combination method

Because, physiologically, cells do not fire with identical profiles to repeated presentations of a stimulus [30], we simulated how the inherent variability of place cells affected their detection by the different methods. We manipulated the reliability of place cell responses by varying the percentage of traversals in which they fired (Fig 3A) and the location of the place field centre, by shifting the position of the activation peak with respect to the average firing field a percentage of the place field width (Fig 3B). The exact traversals that cells fired in were uniformly randomized, while the exact deviation of the place field centre was randomized using a Gaussian probability curve, and was repeated 10 times for each parameter.

The Peak, Stability and Information methods increased their sensitivity as cells became more reliable, reaching a maximum when cells fired in over 30% of traversals. The Combination method also increased in sensitivity with increased reliability, but at a slower rate, reaching a lower maximum sensitivity only when cells fired in 100% of traversals (Fig 3C). Varying reliability did not affect the specificity of any of the methods (Fig 3D). The effect of reliability on the Peak and Information methods comes not from any explicit comparison of cells across traversals, but because unreliability decreases the size of the place field peak in the average fluorescence map, due to the existence of traversals in which the cell is inactive.

Increased variability of the location of the place field centre caused a reduction of sensitivity above 50% variability for the Peak and Combination methods, above 80% variability for the Stability method, and above 120% for the Information method (Fig 3E). Specificity of all methods was again unaffected by changes in variability (Fig 3F).

When changing both the variability and reliability, results were again similar to varying each individually, with maximal sensitivity for all methods occurring when cells had high reliability and low variability (Fig 3G), and stability being similar across all conditions (Fig 3H).

Overall, the Information method had the highest sensitivity, especially across highly variable model cells. However, its specificity was lower across the board than the Peak method, for which the sensitivity was lower at a high variability. This difference is likely due to the respective percentile thresholds (99th for the Peak method and 95th for the Information). The response pattern of the Peak method is arguably preferable in this instance, as highly variable cells which are active in different locations from traversal to traversal do not fit within what is understood to be a place cell. Conversely, the Combination method's sensitivity was decreased by even small decreases in reliability, while the Stability method was more robust than the Peak method to high variability, but less so compared to the Information method, and had low overall specificity.

## Place cell parameters affect methods differently in variable populations

Although the modulation of the various place cell parameters (peak, width, reliability and variability) illustrate the effect of the singular parameters on the performance of the methods, having the same parameter setting for each place cell is unlikely to reflect real place cell populations. Moreover, we only combined 2 subsets of parameters (peak and width, reliability and variability), but not all four. Therefore, to mimic a more realistic model population, we performed a further simulation, where we randomly selected a place field peak, width, variability and reliability for each individual cell of the population (Fig 4A). As a result, each of the

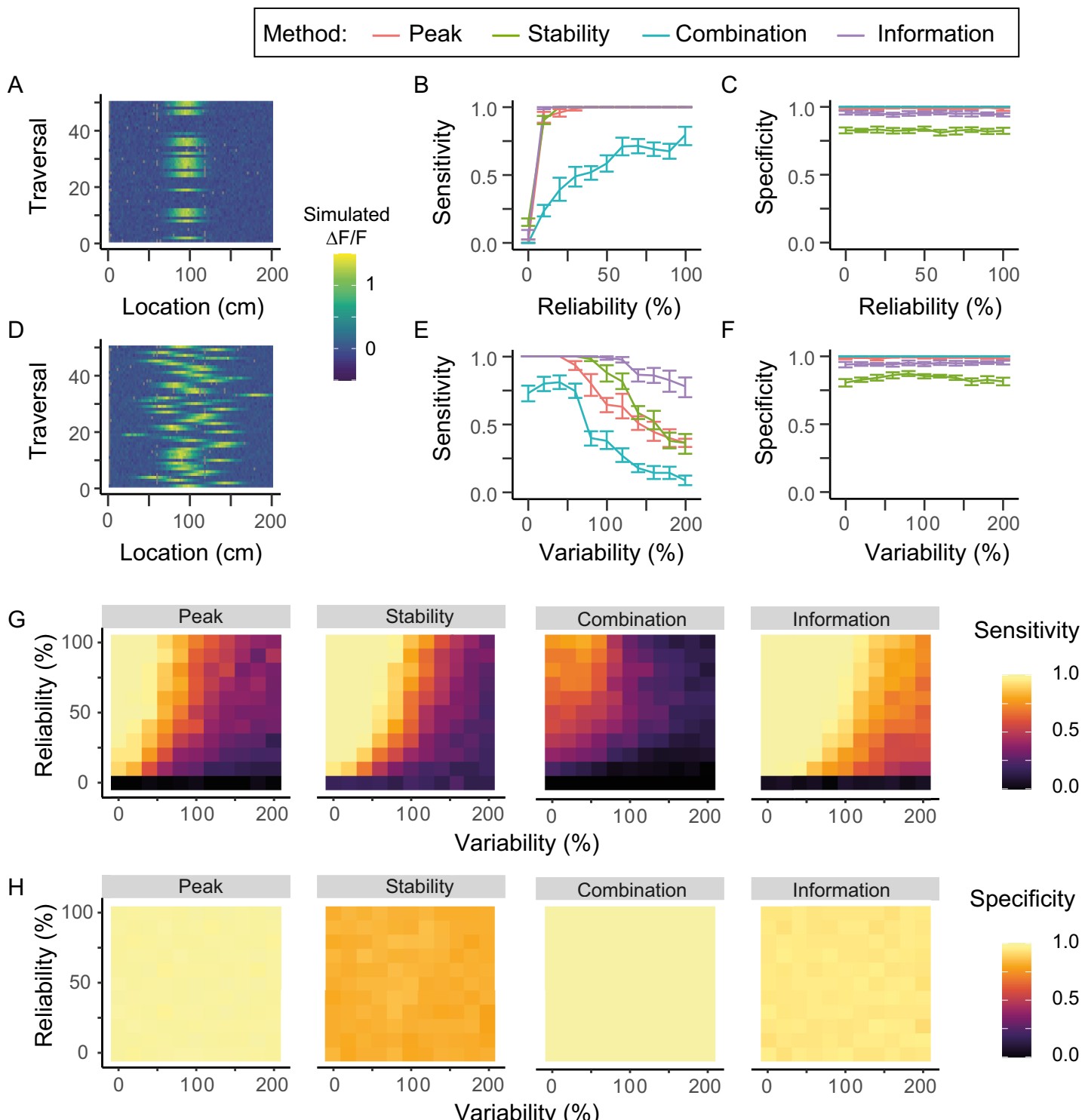

**Fig 3. Place cell identification in variable and unreliable place cells.** (A) Fluorescence map for a place cell with a reliability of 40% (B) Sensitivity of methods as a function of the reliability of a place cell. (C) Specificity of methods as a function of reliability. (D) Fluorescence map for a place cell with 60% variability. (E) Sensitivity of methods as a function of the variability of a cell's place field across different traversals. (F) Specificity of methods as a function of variability. (G) Surface plots of the sensitivity of each method as a function of both reliability and variability. (H) Surface plots of the specificity of each method as a function of both reliability and variability. Grey points in A and D indicate the mouse did was not recorded at that location in that particular traversal, due to the limited frame rate of our recordings. Data shown are the means of 10 randomly created datasets, with the error bars representing 95% confidence intervals.

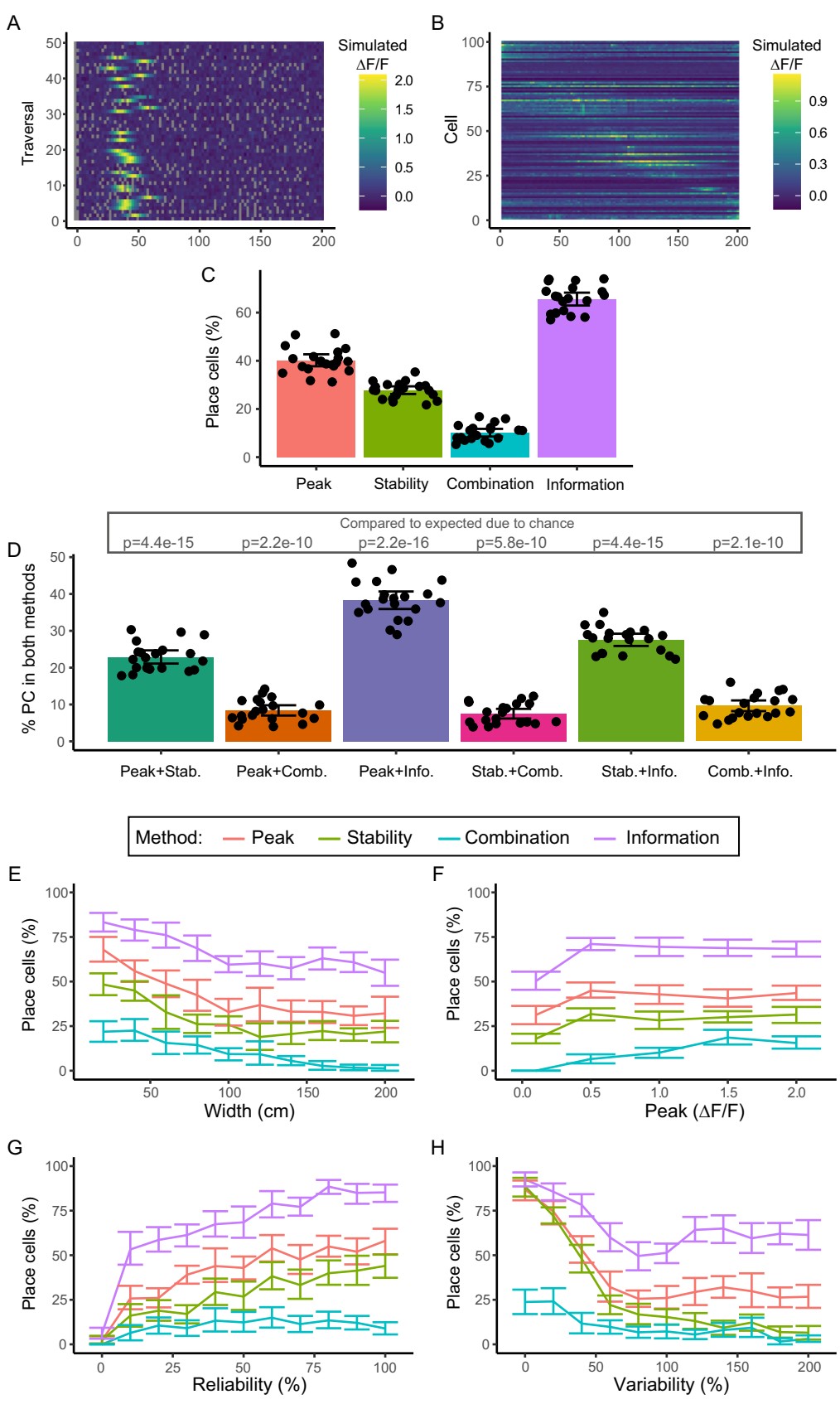

**Fig 4. Place cell identification in modelled populations.** (A) Example place cell with a width of 20 cm, a peak fluorescence of 2, a reliability of 50% and a variability of 40%. (B) Fluorescence maps over location of example population ordered by the location of the highest fluorescence peak in the fluorescence map. (C) Percentage of modelled cells identified as place cells by the methods. (D) Comparison of percentage of modelled cells identified as place cells by two methods. P-values from comparison between observed and expected overlap, using paired t-tests with Holm-Bonferroni correction for multiple comparisons. The percentage of cells classified as place cells by the four methods depending on its (E) width, (F) peak, (G) reliability and (H) variability. For C and D, each data point is a repeat, bars show means, with error bars showing 95% confidence intervals.

100 cells in the population had a different combination of these parameters. In this simulation, we did not include control cells, as we expected enough modelled cells to behave unlike place cells (i.e. those with a low reliability, high variability, low peak and/or high width, Fig 4B). We repeated this simulation 20 times, each time determining the percentage of cells classified as place cells by each of the methods.

We found a significant difference in the number of place cells identified by each of the methods ($p = 1.14 \times 10^{-14}$, repeated measures ANOVA), with the Information method detecting the highest number (65.6 +/- 5.7% (mean +/- std)), followed by the Peak method (40.2 +/- 5.3%) and the Stability method (27.8 +/- 3.4%), while the Combination method only classified 10.2 +/- 3.4% of cells as place cells. Interestingly, despite the Stability method having a high sensitivity and low specificity in the previous simulations, this method classified a lower number of cells as place cells compared to the Peak and Information methods, which previously had a higher specificity. This is likely because the increased variability of the population will cause an increase in the intercell correlation compared to the simulations where the place fields were all strictly in different positions and equidistant from each other. This will therefore cause an increase in the correlation threshold required for cells to be included as place cells, and thus an increase in specificity and decrease in sensitivity, resulting in a lower number of cells included as place cells overall. At the same time, on average the place cells will be less correlated than in our previous simulations, where cells always were highly consistent in all but the manipulated variable, and thus fewer are presumably sufficiently correlated to be classified as place cells (e.g. many will have higher variability and lower reliability so activity will vary from traversal to traversal).

We were interested to see if the same model place cells were identified by the different methods, or if different methods identified independent populations. To this end, we determined the overlap between each of the methods, and compared this to the overlap expected by chance. All combinations of methods had a higher overlap than expected by chance (Fig 4D, $p = 4.4 \times 10^{-15}$, $p = 2.2 \times 10^{-10}$, $p = 2.2 \times 10^{-16}$, $p = 5.8 \times 10^{-10}$, $p = 4.4 \times 10^{-15}$, $p = 2.1 \times 10^{-10}$ for the Peak-Stability, Peak-Combination, Peak-Information, Stability-Combination, Stability-Information and Combination-Information comparisons respectively, paired t-tests with Holm-Bonferroni correction for multiple comparisons), indicating that an overlapping population of cells were identified as place cells by the different methods.

Lastly, we examined how the various place field parameters affected the percentage of cells identified by the methods. For each population, we determined the percentage of cells with a particular value for each parameter (e.g. all cells with a width of 100 cm, regardless of the values for the other parameters) that were identified as place cells. This illustrates again the difference in the percentage of cells identified by each method (Fig 4E–4H), but shows that the effect of the parameters is largely the same across the methods (e.g. increasing place field width decreases the number of place cells identified by each method). In addition, the way each parameter affected the percentage of place cells showed a similar trend to the effect of these parameters on the sensitivity in the previous model (Figs 2 and 3), though the maximum percentage of place cells detected is lower in each case than the suggested by the sensitivity. This is

presumably due to the increased sources of variation in the place cell activity in this simulation: a cell with optimal reliability is likely to be sub-optimal in the other parameters. The notable exception to this is the Combination method, which previously only identified cells as place cells when their field width was within a window between 20 and 120 cm (Fig 2C). In this latest simulation, however, no such defined window exists and, like the other methods, the number of cells decreases gradually as place field width decreases. This is likely because a place cell with a small place field but a high variability can still be classified as a place cell by the Combination method, as increased variability will cause the place field in the average fluorescence map to seem wider, as illustrated in S3M Fig, so that it falls within the window for detection.

## The information method detects the most place cells in real datasets

The performance of the different methods of place cell detection on model data predicts that, on real datasets, the Peak method will have the highest sensitivity and specificity, with the Combination method being less sensitive and the Stability method often being less specific. This would suggest that in a real dataset, the Combination method would identify fewer place cells than the other two methods, while the Stability method, because it has a higher false positive rate on model data, might (inaccurately) detect more cells than the Peak method.

To test this, we collected neuronal calcium data from the stratum pyramidale of dorsal CA1 in 4 mice (8 imaging sessions in total) traversing through a 1D virtual reality corridor of 200 cm (Fig 5A). We applied each of the four methods to these data to identify which cells could be categorised as place cells (Fig 5B). There was a highly significant difference in the percentage of cells classed as place cells by the different methods ($p = 1.35 \times 10^{-9}$, linear mixed-effects model with method as a fixed-effect and mouse ID, with session number nested, as a random effect), with the Information method detecting the highest number (47.0 +/- 12.2% (mean +/- std)), followed by the Peak method (34.5 +/- 11.1%) and the Stability method (16.5+/- 8.8%), while the Combination method only classified 3.9 +/- 1.9% of cells as place cells. The properties of these cells can be seen in an example recording showing the fluorescence maps of all cells, ordered by location (Fig 5C). In addition, the populations of cells identified by each of the methods are shown (Fig 5D–5G). Although no ground-truth exists for what should be included as a place cell, most identified cells seem to have place fields, suggesting classifications are largely appropriate.

## Differences between detection in modelled and real data

Based on the results from our modelled datasets, we expected the Stability method to classify more cells in a real dataset as place cells than the Peak or Information methods, but in fact it identified significantly fewer ($p = 0.0012$ and $p < 0.0001$ respectively, Tukey post-hoc comparison). We probed the reasons for this difference both by modulating the properties of the modelled cells and inspecting examples of real cells identified differently by the different methods. Because the Stability method compares the activity of each cell to that of other cells in the population, while the Peak and Information methods compare each cell to a shifted version of themselves, the properties of non-place cells could preferentially affect the performance of the Stability method. We therefore modulated the number and firing properties of the modelled non-place cells to see if this reduced the number of cells identified by the Stability method. Increasing the percentage of place cells and adding random calcium peaks to the non-place cells resulted in an increase in specificity (S7A–S7F Fig), likely because it decreased the correlation of the fluorescence maps of the first and second halves of the sessions for the non-place cells. In addition, we tested the effect of having a population with fully overlapping place fields, which again caused an increase in specificity (S7G–S7I Fig). Lastly, adding in random calcium

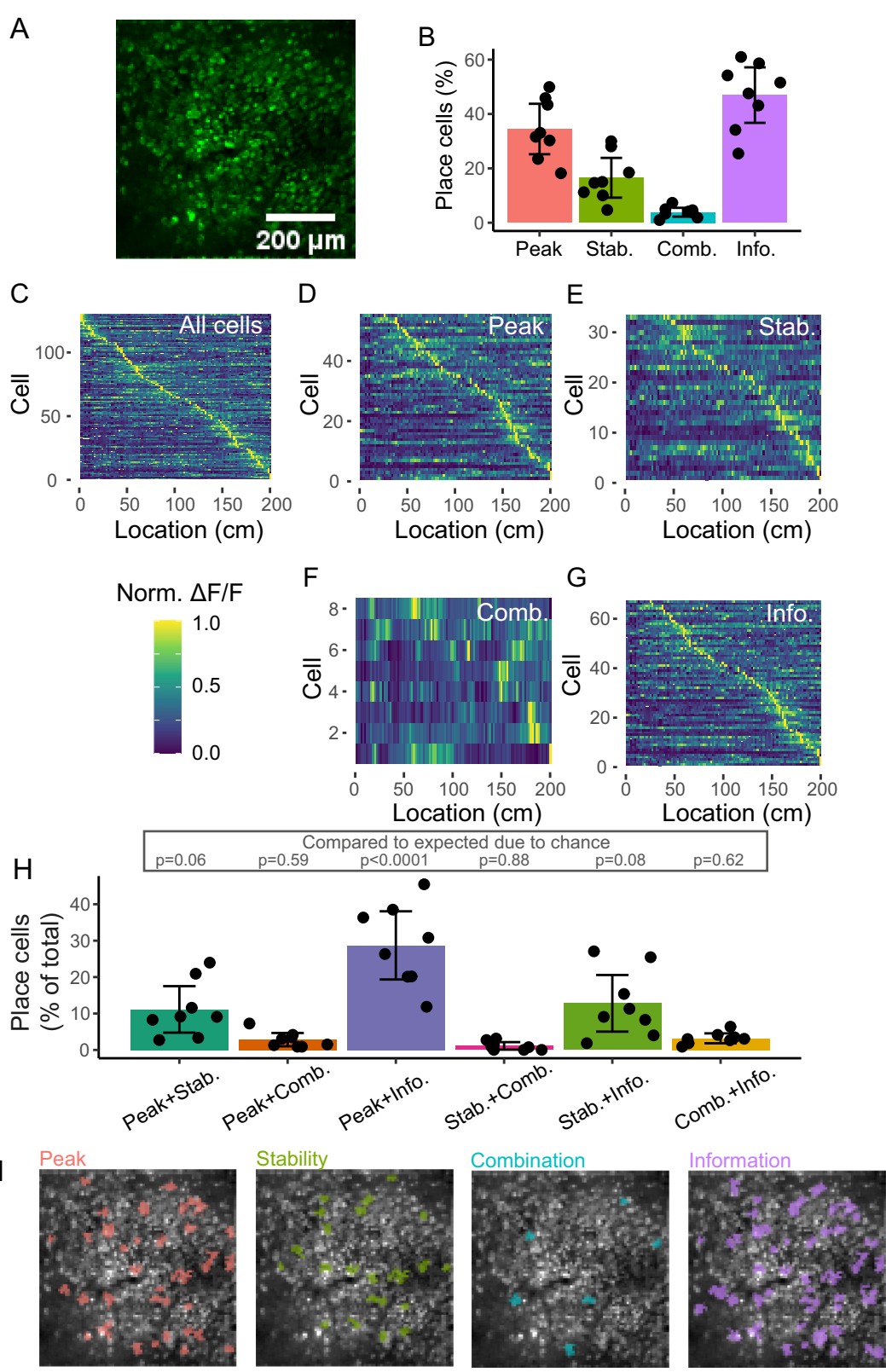

**Fig 5. Place cell identification in real datasets.** (A) Two-photon recording of GCaMP6f in pyramidal cells. Image is a standard deviation Z projection of a single recording. (B) Percentage of ROIs identified as place cells by the methods. (C) Fluorescence maps over location of all ROIs in an example recording. Cells in this recording identified as place cells by (D) Peak, (E) Stability, (F) Combination and (G) Information methods. (H) Comparison of percentage of ROIs identified as place cells by two methods. P-values from comparison between observed and expected overlap, using paired t-tests with Holm-Bonferroni correction for multiple comparisons. (I) Two photon image from A showing ROIs identified as place cells by the four methods. For B and H, each data point is a recording, bars show means, with error bars showing 95% confidence intervals.

peaks to the place cells in addition to the non-place cells caused a similar increase in specificity for the Stability as adding them only to non-place cells (S7J–S7L Fig). Although these factors decrease the overall number of place cells detected by the Stability method, due to the reduction of false positives, this could not wholly account for the observed decrease in cells detected in real data compared to the other methods.

Visual inspection of the firing properties of cells identified by the Peak and Information methods, but not by the Stability method, however, showed that the difference in detection is caused largely by cells that have different place fields at different times in the session (S8 Fig), in addition to cases where cells either gain or lose a place field during the session. The Stability method fails to identify these cells, because their mean activity over location for the first half of the session is different from the second. This type of structured variation in the place field location was not captured by our simulations of place field activity, because reliability and variability were varied randomly, which did not affect the ability of the Stability method to detect place cells. Minor changes to the Stability method or experimental design should prevent this effect, for example by comparing odd to even traversals, or by adding a familiarisation period for the animals at the start of each session which could ensure place fields are more stable when the experiment begins. In addition, a paradigm where the animals repeatedly enter the environment, as is the case in our experiment because traversals are separated by a tunnel, may induce switching between concurrent spatial maps [27], thus decreasing the overall stability of the cells. However, it is an open question whether some of the cells identified by the Peak and Information methods should really be classified as place cells, for example those with unstable place fields over the session.

## Different populations of real CA1 pyramidal cells are identified as place cells by the different methods

To understand whether the different methods simply identify more or fewer of the same population of cells as place cells, we determined the percentage of all ROIs that were identified by two methods. Only 1.1% of cells were classed as place cells by all four methods. 11.1 +/- 7.6% of ROIs were identified as place cells by both the Peak and Stability methods (Fig 5H), while only 2.8 +/- 2.2% of ROIs were identified as place cells by the Peak and Combination methods. The Peak method showed a large overlap with the Information method, with 28.7 +/- 11.2% of cells identified as place cells in both methods. Only 1.1 +/- 1.2% of cells were classed as place cells in both the Stability and Combination methods, while 12.8 +/- 9.3% of cells were identified as place cells in both the Stability and Information methods. The information and Combination methods identified 3.2 +/- 1.6% as place cells. These overlaps were significantly different between pairs of methods, when corrected for expected overlap given the different percentages of cells identified by each method (p = 0.0014, linear mixed-effects model with method as a fixed-effect, expected overlap as a covariate and mouse ID, with session number nested, as random effects). This significant difference is due to a higher overlap between the Peak and Information methods compared to the other combinations of methods (p = 0.075,

p = 0.01, p = 0.003, p = 0.002, p = 0.004 compared to Peak-Stability, Peak-Combination, Stability-Combination, Stability-Information and Combination-Information respectively, Tukey post-hoc comparison). Indeed, we only observed a significant different overlap from the expected overlap by chance for the Peak-Information comparison (p<0.0001, Tukey post-hoc comparison), but not for any of the other comparisons (p = 0.059, p = 0.59, p = 0.88, p = 0.084 and p = 0.62 for the Peak-Stability, Peak-Combination, Stability-Combination, Stability-Information and Combination-Information comparisons respectively) This suggests that the Peak and Information methods identify an overlapping population of cells, but this population is independent of the cells indentified by the other two methods. These results are notably different from our findings in the simulated populations (Fig 4), where all methods identified an overlapping population of cells. This suggests that additional variation is present in the real data that we did not capture in our model datasets, but that influences place cell identification by the different methods. Such variation could include out of field firing or structured variations in the place field over a session, as demonstrated in S8 Fig.

## Identified place cell populations differ on key characteristics

To better understand the nature of the different populations of cells identified by each method, we characterised their properties, comparing basic physical features, and functional characteristics. Place cells and non-place cells identified by the different methods were of similar sizes (Fig 6A; a smaller size could indicate a noisier image and a higher chance that the automated cell detection by Suite2P had made a mistake as identifying it as a cell). Place cells and non-place cells identified by all methods were also similarly active, as assessed by both mean and maximum spiking rate (Fig 6B and 6C, calculated from the fluorescence profile [31]).

To investigate the information about location carried in place or non-place cells identified by the different methods, we calculated the mutual information between the cell's activity and the animal's location, providing an approximation of the spatial information carried in each cell's calcium signals [32–34]. More spatial information was carried in place cells identified by the Peak and Information method compared to non-place cells, but there was no difference in mutual information held by the place cell and non-place cell populations for the Stability or Combination methods (Fig 6D). This is likely because cells containing spatial information that the Peak and Information methods would call place cells are classified as non-place cells using the Stability or Combination method. However, even the cells classified as non-place cells in the Peak and Information method contain some level of spatial information. Indeed, a previous study found that cells may convey information about location without being defined as a place cell [35]. For example, a cell whose activity ramps up gradually as the mouse progresses through the environment does not have a clear place field and will not be included as such in these methods but does communicate substantial spatial information.

Place cells exhibit no particular spatial organisation [11], and any such organisation compared to non-place cells might indicate an unwanted selection by one of the methods, e.g. based on illumination levels in one part of the field of view. Therefore, we determined the physical location of the detected place cells within the field of view of the recording. We expect the place cells and non-place cells to be randomly distributed within the field of view, as previously reported [11]. We determined the location of the centroid of each cell, labelled them as either place cell or non-place cell according to the different methods and then determined the distance from each place cell to all other place cells, and each place cell to all other non-place cells (Fig 6E). Place cells were no more similar in location to other place cells than to non-place cells for any of the methods, suggesting that place cell detection is not being influenced by position within the field of view.

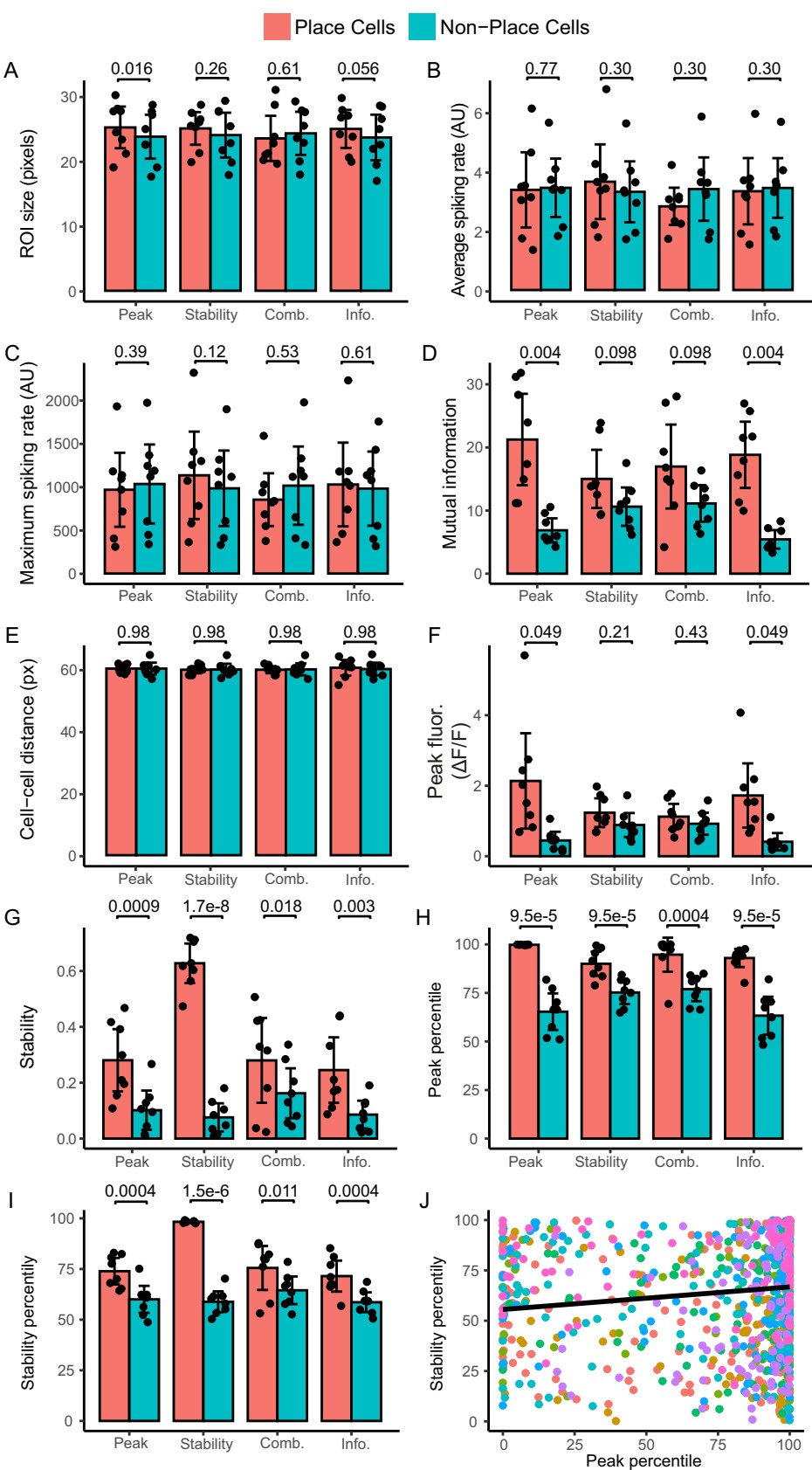

**Fig 6. Characteristics of place cell populations.** (A) The ROI size, as a percentage of the size of the largest ROI in that recording, of the place cell and non-place cell population in the four methods. (B) Mean spiking rate of place cells and non-place cells. (C) Maximum spiking rate of place cells and non-place cells. (D) Mutual information between cell activity and location for each classification by the four methods. (E) Mean distance between the centroids of the place cells (pink), and the distance from place cells to non-place cells (blue). (F) Peak intensity in the fluorescence maps, the main characteristic used in the Peak method. (G) Intrasession stability, measured using the Pearson correlation coefficient, between the fluorescence map of the 1$^{st}$ half and the fluorescence map of the 2$^{nd}$ half for place cell and non-place cell, the main characteristic used in the Stability method. Percentile of (H) peak intensity and (I) intrasession stability compared to shuffled controls. (J) Correlation between percentile of peak intensity and intrasession stability for all cells. In J, each colour indicates a single dataset. For the other figures, bars are the means of 8 datasets, black dots are values for each individual dataset and error bars represent the 95% confidence interval. P-values for A-I are from multiple paired t-tests with Bonferroni-Holm correction.

## Dependence of place cell detection on model parameters

The lack of overlap between cells identified by the different methods suggests that they identify fundamentally different populations of cells as being place cells, i.e. there is not a single population of cells that fits all the different definitions of place cells used by the different methods. To explicitly test for this, we calculated the extent to which the parameters which underlie classification for the Peak and Stability (peak intensity and correlation in firing across the session, respectively) were represented in the groups of place cells that were selected by each other method (The Combination method uses several parameters so could not be analysed in this way, and the parameter for the Information method was already compared by testing the mutual information). The Peak and Information methods had significantly higher peak intensity in the place cell group compared to the non-place cells (Fig 6F), while there was no difference between the place and non-place cells for the Stability and Combination methods.

Place cells identified by all methods had an increased correlation in activity across the session in place cells compared to non-place cells–i.e. they were more stable (Fig 6G). Thus, the Peak, Combination and Information methods also select for stable cells–likely because stable place cells are consistently active in the same location across traversals, yielding on average a robust peak, which in turns leads to increased spatial information.

Though both the peak intensity and stability are parameters used as a measure to identify place cells in their respective methods, the Peak method compares the values of each individual cell to the peaks of shuffled versions of that same cell, while the Stability method compares the correlation to randomly selected other cells, rather than comparing the values to an absolute threshold. This means that place cells are not necessarily the cells with the highest absolute peaks or highest correlation, but rather the highest value relative to their own shuffled controls. Therefore, we calculated the percentile these values fell in, relative to the shuffled controls, for each cell. Place cells identified by all methods had an increased percentile score for both the stability and the peak parameters (Fig 6H and 6I).

To understand why the methods select place cell populations with similar peak intensity and stability, we determined to what extent the parameters used to define place cells were independent from each other. We correlated the percentiles for the peak intensity and intra-session correlation for each cell across all datasets (Fig 6J). There was a small but significant correlation between the percentiles for peak fluorescence and intra-session correlation (Pearson correlation coefficient, R = 0.13, p = 3.7x10$^{-5}$). Although this suggests the two measures are not completely independent–cells with high peak values have a tendency to have more highly-correlated firing patterns within a session–we did not see this reflected in the overlap between the methods (Figs 4H and S7H). This may be because the correlation coefficient between peak intensity and intra-session correlation is too low to have an impact. In addition, it may be the

case that the more correlated cells are not the same as those that reach the threshold for inclusion as a place cell in one or both methods, so are not reflected in the place cell overlap.

## Optimisation of the combination method

The Combination method identified many fewer cells in our data than in previously publications [11,20] or the other two methods applied here. We wondered if this could be due to the number of thresholds the Combination method applies, ([11,20] code to implement the Combination method shared by the authors), requiring putative place cells to fall within specific parameter ranges. These were presumably optimised for the published datasets and may make the method less transferable to other datasets. To test this, we investigated how varying these parameters and thresholds affected place cell detection in our model and experimental datasets (S9 Fig; Methods). Sensitivity for detecting model place cells could be improved by increasing the threshold for place field fluorescence compared to baseline (S9B Fig) or reducing the threshold in-field/out-field fluorescence ratio to 2 (S9E Fig), but neither modification caused a substantial increase in the number of cells classed as place cells in experimental data (3.9%; S9G Fig). Instead, the critical feature of the Combination method is the bootstrapping method, which splits the fluorescence data into chunks and shuffles them. Cells are only classed as place cells if fewer than 5% of each cell's shuffles also fit the classification criteria. If we replaced the bootstrapping methods from the Combination method with that of the Peak method, which instead offsets fluorescence traces relative to time, causing more disruption to the association between fluorescence and location, we identified 14.8% of ROIs in our experimental data as place cells (S9G Fig), which is within the published range using the Combination method [11], though below the numbers identified by the Peak or Information methods in our data (Fig 5B). Some of these newly identified cells now overlapped with those identified by other methods, 9.3% of all ROIs now overlapping with cells identified by the Peak method, 5.6% overlapping with those identified with the Stability method (S9H Fig), and 11.7% with the Information method. Therefore, when using the new bootstrapping method, around 75% of PCs identified by the Combination method were also identified by the Peak method. However, the levels of overlap between the Combination method and the other methods were still as expected given the frequency of detection by each method (p = 0.17, p = 0.30, p = 0.10 for the Peak-Combination, Stability-Combination and Combination-Information comparisons respectively, Tukey post-hoc comparison following a linear mixed-effects model with method as a fixed-effect and mouse ID, with session number nested, as a random effect). This means that the cells identified by each method are still likely to be from independent populations (i.e. if a cell is classified as a place cell by one method that does not increase its chance of being classified as a place cell by another method).

## Discussion

We compared four different methods for detecting place cells in two-photon calcium imaging data from mice running through a one-dimensional environment, comparing sensitivity and specificity of the different methods to model cell data, and whether they identified the same real populations of cells as place cells. The methods performed very differently at detecting model place cells and were differentially sensitive to changing parameters (e.g. number of traversals of the environment, place cell width, consistency of firing). This suggested that variability in firing properties across real hippocampal CA1 pyramidal cells would lead to different cells being classified as place cells by the different methods. This proved to be the case: there was very little overlap in real CA1 place cell populations classified by the different methods, except for the Peak and Information methods, which largely overlapped, because the key

features used by the different methods to classify place cells proved only weakly correlated. When research groups use these different methods to detect place cells, they therefore identify and study largely independent populations of cells. Researchers should therefore explicitly consider the properties they are selecting for when choosing a place cell detection method, as this decision will determine which cells they do and do not select. In the future, use of a single method across different groups would better allow comparison of results across the field. The increased selectivity and specificity of both the Peak and Information method when using model data, leads us to conclude that these methods will usually be the best for detecting place cells in calcium imaging data. However, compared to the Information method, the Peak method does not inherently make assumptions regarding the spatial information held by potential place (or non-place) cells, and this neutrality to the information contained within the activity better reflects the capacity of cells without a clear location peak to contain spatial information [35]. Finally, the Peak method has previously been successfully employed to detect place cells in electrophysiological recordings [24], making it widely applicable beyond two-photon recordings. Therefore, we believe the Peak method will usually be the preferred method for place cell classification. Nevertheless, some curation of detected cells may still be desirable, as some cells identified by the Peak method showed unstable firing patterns that may be considered too dissimilar to the firing properties of a traditional place cell.

## What is a place cell?

Because our results reveal that the choice of detection method will determine the population of cells identified as place cells by a given experiment, it is important to consider what the "desirable" properties of a place cell are, and whether the different classification methods detect these properties. O'Keefe's first description of place cells defined them as "those for which the rat's position on the maze was a necessary condition for maximal unit firing." [1]. Of the four methods we assess here, the Peak method comes perhaps closest to this original definition, requiring cells to have high and consistent firing in one location compared to others, by statistically comparing whether the peak response consistently occurs at the same location. The Combination method achieves something similar by requiring firing to be above a certain threshold for a defined range of place field sizes, while the Stability method only requires the firing pattern to be stable over location and time without the explicit requirement for the peak to be highest in one location. The Information method uses the cells' spatial information, which does capture the definition, but also makes the assumption that place cells always have an increased amount of spatial information, which is not necessarily the case [35]. For example, cells' activity could ramp over space, thus containing spatial information without having a clear peak in activity in a given location.

The reason for the divergence of place cell definitions from the initial more constrained description is the increased emergence of variability in "place cell" properties. We now know that the size, shape, number and stability of place fields are not constant but can be affected by the environment and where in CA1 they are recorded: Place fields are wider, and more numerous, in larger environments, more stable in the presence of cues and smaller and less spatially selective in dorsal vs. medial CA1 [36–40]. Clearly methods that discriminate place from non-place cells according to a rigid interpretation of the "classic" definition will miss cells with these more variable properties. Indeed, we found that the Combination method has a lower sensitivity to detect cells with a place field outside a narrow range (100–120 cm), and failed to detect place cells as variability of place field position increased or reliability of firing reduced. Thus, we predict that experimental manipulations that alter the size of the environment and number of cues provided would reduce the number of place cells identified using the Combination method, but not using the Peak method.

Similar changes in place cell properties can occur due to the method of acquiring neural data, impacting on which might be the most appropriate method to choose to detect these cells. Many experiments that record the activity of large populations of cells, either using calcium imaging, as here, or multi-channel electrode arrays are conducted in head-restrained mice, while the animals are running on a 1D linear track [11,41–43]. These conditions likely cause place cells in our experiments to have broader place fields than in real world environments [15,44], and to be more directionally-sensitive, due to the 1D nature of our linear track [11,45]. The place cell detection method used for such imaging data therefore needs to recognise these less sharply tuned activity patterns. Both the Peak and Information methods found larger numbers of place cells in the experimental data, and selected cells which were both relatively stable and spatially selective. This increased number selected as place cells in our head-restrained imaging data occurred perhaps because of their broader spatial selectivity compared to classic tetrode recordings, suggesting that similar studies using head-fixed mice and linear tracks would also miss spatially-selective cells if a more conservative method were employed.

Less controllable variations in experiments between labs, for example the amount the mice run in an experiment, will also differentially affect the number of cells classed as place cells by the different methods. Our simulations suggested that all four methods would be more sensitive at detecting place cells as mice made more traversals of the environment, but the Combination method would miss cells with even relatively high numbers of traversals (missing over 50% of cells when 10 traversals were made). Furthermore, the Stability method increasingly falsely characterised non-place cells as place cells as the number of traversals increased, because non-place cells had higher intracell correlations as the number of traversals increased. Thus, only the Peak and Information methods could correctly identify place cells with low or high numbers of traversals. Choice of place cell categorisation method will therefore clearly affect the number of cells identified in experiments where animals run to different degrees. Crucially, however, the confounding of place cell identification with running behaviour in 2 of the 3 methods could lead to inaccurate conclusions about how experimental manipulations affect place coding. If, for example, mice of a certain genotype run less, generating fewer traversals, they could appear to have fewer place cells using the Combination method, or more place cells if using the Stability method, without the number of "true" place cells having changed.

In this study, we only investigated the effect of varying the number of traversals on place cell detection, but other characteristics of the locomotion may also have an effect, for example running speed or the amount of starting and stopping. As these effects may interact with the environment or task, (for example, mice slowing down before a reward delivery), we suggest testing the performance of a chosen method of place cell detection on a model dataset with the expected locomotion characteristics before using it to detect place cells experimentally. We provide source code (https://github.com/DoriMG/place_cell_methods) to facilitate this process, into which experiment-specific patterns of locomotion can be loaded and used to predict place cell detection using the four methods described here.

Lastly, in our simulations we modelled each cell independently. By modelling our cells this way, we are potentially missing out on network dynamics within the population, where cells fire dependently on each other, or are synchronously affected by a third factor, such as global theta oscillations. However, because each cell is individually evaluated in the Peak, Combination and Information methods, these methods will not be affected by such dynamics, and as such adding network dynamics to our models would not affect these results. In the case of the Stability method, the cells are compared to shuffles across the population, and both the sensitivity and the specificity of this method will very likely be affected by network dynamics. An increased correlation across the population will cause an increase in the shuffled correlation,

thus a higher threshold for intracell correlation for cells to be classified as a place cell. If the increase in threshold is large enough, the sensitivity of the Stability method will therefore decrease, classifying fewer place cells, while increasing the specificity, meaning fewer non-place cells will be falsely marked as place cells. Indeed, we saw a similar effect on specificity for the Stability method when all place fields in the population were in the same location, causing an increased correlation across the population (S7I Fig).

In addition to correlations of activity across the population of pyramidal cells, network dynamics may also cause the activity of a cell in a given traversal to depend on its own prior activity (e.g. if a cell fires in traversal n, it might be more likely to fire in traversal n+1). Such an intracell dependence will affect the Peak, Combination and Information methods in the same way as they are affected by altering the reliability (i.e. how much activity varies across location across all traversals), as these methods classify place cells based on the average fluorescence map across all traversals, and do not take individual traversals into account. However, the Stability method may be affected if such dependency between traversals causes a cell to lose or gain a place field, or if the place field shifts position between the first and second half of the session, as demonstrated in S8 Fig. Therefore, if one expects strong network dynamics to be present in a population of place cells, we recommend using caution when employing the Stability method.

In conclusion, because place cells have a more variable activity pattern than was originally thought, particularly across the large populations and range of experimental paradigms permitted by calcium imaging, classification methods should be sufficiently able to identify cells that vary in terms of the size, and reliability and variability of their place field location. However, the different methods we tested select largely different populations of cells which differ in key characteristics, highlighting that choice of place cell classification method is critical for the conclusions a study will draw as to the nature of place cells. We provide model place cell code to help researchers test their how their chosen methods or experimental manipulations might affect detection and the properties of place cells in their data. However, we suggest that consensus in the field for an identification method would help inter-study comparability. Overall, we found that the Peak method demonstrated the optimal high selectivity and specificity for selecting model place cells that was robust to moderate changes in place field properties but decreased appropriately as the reliability and variability for the place field decreased. It detected many place cells in a real dataset and these cells carried more mutual information about location than non-place cells, but without the concerns of false positives of the Information method. The Peak method has previously been successfully applied to electrophysiological recordings [28], and we therefore would predict that the properties of place cells in imaging data detected using the Peak method will most accurately reflect the properties of place cells detected in a similar manner in electrophysiological studies. Because of its simplicity and lack of assumptions about the spatial information held by place and non-place cells, for most experimental designs we would therefore recommend use of the Peak method for classifying place cells in calcium imaging data, but advise that whichever method is chosen, experimenters consider the likely impact of that method choice on the cells identified.

## Materials and methods

### Ethics statement

Experiments were approved by the UK Home Office, in accordance with the 1986 Animal (Scientific Procedures) Act as well as the University of Sussex Animal Welfare Ethical Review Board.

## Animals

Experiments used four C57/BL6 mice (2 female, 2 male) expressing the genetically-encoded calcium indicator GCaMP6f under the control of a Thy-1 promoter (C57BL/6J-Tg(Thy1-G-CaMP6f)GP5.5Dkim/J). The mice were housed in a 12h reverse dark/light cycle environment at a temperature of 22˚C and were given ad libitum access to food and water.

## Hippocampal cranial window surgery

Surgery was performed when mice were a minimum age of eight weeks. Before surgery, mice received subcutaneous injections of dexamethasone (60 μL, 2mg/mL), saline (400 μL) and buprenorphine (40 μL, 0.3mg/mL diluted 1:10 in saline) to reduce inflammation, for hydration, and pain relief respectively. Mice were maintained at 0.8–2.0% isofluorane anaesthesia for the duration of the surgery. Body temperature was maintained at 37˚C using a homeothermic blanket (PhysioSuite, Kent Scientific Corporation). A craniotomy was inserted above dorsal CA1 as previously described [29]. Briefly, the skin above the skull was removed and the skull was scored to increase the surface area for binding dental cement. A custom-made stainless steel headplate was then fixed to the skull with black dental cement (Unifast Powder mixed with black ink (1:15 w/w) and Unifast Liquid). A 3 mm diameter craniotomy was then performed 2mm posterior to bregma and 1.5 mm lateral to the sagittal suture. Following the removal of the skull flap and the dura, brain tissue overlying the hippocampus was aspirated (New Askir 30, CA-MI Srl) until vertical striations of the corpus callosum were visible. We then inserted a custom 3D printed cannula (2.4 mm ID, 3 mm OD, 1.5 mm height) made of a biocompatible Dental SG resin (FormLabs) so that the glass coverslip at the bottom of the cannula lay directly on top of the brain tissue. The top of the cannula had a rim (0.2mm height, 3 mm OD) resting on top of the skull, which was attached using tissue adhesive (3M VetBond) and then covered with more dental cement. A rubber ring was then attached on top of the headplate for subsequent use as a well for the water needed for the water-immersion microscope objective. The mice were given an injection of meloxicam (125 μL, 5 mg/ml) as an analgesic near the end of the surgery and then received meloxicam (200 μL, 1.5 mg/mL) for 4 days following the surgery via oral admission. Their health was monitored and they were weighed daily.

## Two-photon imaging

**Habituation.** A week or more after surgery, the mouse was habituated to the imaging rig by head-fixing it for an increasing amount of time each day for at least a week before it was imaged. During the habituation it was also presented with the virtual reality environment several times to make it familiar with this setup.

**Imaging rig (S2 Fig).** The mice were head-fixed above a polystyrene cylinder, on which they could run. The cylinder was fitted with a rotary encoder (Kübler, 4096 pulses per revolution). Two screens in front of the mice were used to display a custom virtual reality (VR) environment designed using ViRMEn [46].

**Stimulus presentation.** The virtual reality environment presented to the mice was a wide corridor, 200 cm long and 80 cm wide, with patterned walls (30 cm high) and floor (see S2 Fig). Three sets of objects (spatial cues) were present outside the walls: blue square pillars with white stripes, light blue cones with white and gray diagonal stripes and grey cylinders with green dots. The objects were placed at 65 cm, 140 cm and 200 cm from the start of the corridor. All objects were 100 cm high and visible from the start of the arena. Both before and after the wide corridor was a dark grey tunnel with a diameter of 30 cm, 50 cm long before the corridor and 45 cm long after the corridor) which served to allow smooth transitions between multiple

presentations of the environment. The mice were not required to perform a task while in the virtual environment.

**Data acquisition.**   The stratum pyramidale of dorsal CA1 was imaged using a two-photon microscope (Scientifica) with a water-immersion objective (CFl75 LWD 16X W, Nikon; 0.80 numerical aperture, 3 mm working distance). GCaMP6f was excited using a Chameleon Vision II Ti:Sapphire laser (Coherent) at a wavelength of 940nm with a gallium arsenide phosphide photomultiplier tube. We used the ScanImage software (Vidrio Technologies, MATLAB) to control the microscope and collect data. The stratum pyramidale was identified from the presence of densely packed cell bodies. Image acquisition used a wide field-of-view (547 x 547 μm) at a low resolution to optimise the acquisition rate (128x128 pixels, 7.51Hz, pixel size 4.27 μm). Sessions lasted between 44 and 45 minutes.

## Image analysis

**Preprocessing.**   Preprocessing was conducted using Suite2P software [31]. Firstly, images were registered using the default settings, then regions of interest (ROIs) corresponding to pyramidal cell bodies were identified based on their morphology (having a diameter of approximately 2 pixels/8.5 μm) and a tau, the decay time for the calcium indicator, of 0.8. We trained a classifier by manually curating the detected ROIs based on the mean image of the original recording, the shape and the activity pattern of the ROI. On average 58 +/- 4% of ROIs were excluded per imaging session. We obtained the calcium signal corrected for neuropil activity for each ROI from the Suite2P output.

For each ROI, fluorescence time courses were normalised to baseline fluorescence by dividing the whole trace by the average intensity in that ROI during the first 100 frames of the recording. For calculations of fluorescence maps, any frames where the mouse was stationary were excluded (defined as the speed being below 10% of the maximum speed).

An extra pre-processing step was used in Dombeck et al. [11] and associated papers, described in full in [12], so we at first used this step as well when replicating this method (hereafter referred to as the Combination method). After ROI extraction and normalisation the whole time series was divided by the baseline, which was defined as the 8th percentile of values in each ~15 second interval. Significant transients were then identified as calcium events that started when fluorescence deviated more than 2 standard deviations from the baseline and ended when the fluorescence returned to less than 0.5 standard deviations from the baseline. Fluorescence outside of the significant events was set to 0. However, we noticed that this pre-processing step led to place cell activity appearing negative, and thus these cells being rejected, if their activity was shorter than 15s and occurred in the presence of a negative baseline. For our data, this led to all cells being rejected at this stage. We therefore did not use this preprocessing method in our analyses.

Subsequently, to test whether the lack of preprocessing explained the low number of cells identified using this method, we amended the preprocessing method to correct for slow drifts in the baseline while preventing division by a negative baseline, by subtracting rather than dividing by the baseline. This caused a significant increase in the sensitivity of the Combination method compared to non-preprocessed traces (S7A Fig).

## Model data generation

We generated model place cells to test the performance of the place cell detection methods. Time series of calcium responses of model cells were generated using real locomotion traces of mice running through a linear virtual environment. 8 datasets containing 184 traversals were split into separate traversals and model locomotion traces were generated by randomly

selecting the required number of traversals (each traversal could be selected more than once). Model place cell properties (described below) were convolved with these locomotion traces to generate a fluorescence map for each cell, with the same "frame rate" as used in the real imaging sessions used to acquire locomotion (7.51 Hz).

For each dataset, we included 20 model place cells–in which fluorescence was modulated by spatial location–and 80 non-place cells–whose fluorescence was independent of location. This ratio was chosen to mimic the percentage of place cells typically reported in experiments using a one-dimensional track and two-photon microscopy [11]. The inclusion of non-place cells was crucial as some of the place cell detection methods use comparisons with other cells in their definition, and thus rely on the population containing non-place cells as well as place cells.

For the model place cells each place field was modelled as a 1D Gaussian field centred at equidistant locations covering the entire environment, mimicking previously reported place cell populations [11,41] (Fig 2A). The sigma of the Gaussian was set to 12.5 cm, such that 95% of values– 4 times sigma–fall within the place field width of 50 cm, the average place field width reported by Dombeck et al. [11]. The peak of the Gaussian (1.3) was determined using the top 10% of each cell's fluorescence in its fluorescence map from 992 cells from 8 datasets.

The noise in all cells was modelled using a Poisson distribution (generated using the MATLAB poissrnd function), with lambda estimated from the raw traces of 992 cells from 8 datasets. The median lambda of the 10% of cells that had the best Poisson fit (235.1, as estimated from the sum of squared errors; SSE) was used as the lambda for our model noise. Raw noise was generated using this parameter, then $\Delta F/F$ of this noise was calculated to obtain the normalised noise used in the model cells. The average $\Delta F/F$ of the noise, as tested on 10000 traces each with a length of 1000, was 0.0024 +- 0.0467, which is 0.18% of the peak value.

Non-place cell traces usually consisted of only noise (Fig 2B, top 80 cells), generated as above, while the noise was added to the generated place field for the place cells (Fig 2B, bottom 20 cells). For S7 Fig, we also considered the impact of non-spatial firing of non-place cells. In our real data, there were on average 0.0062 peaks per frame, i.e. a calcium peaks occurred about every 160 frames. We therefore randomly added a range of peaks to non-place cell traces, from 0–0.02 peaks per frame.

## Manipulation of place cells

We manipulated the place cell properties in order to model the imperfect nature of real place cells. We varied the place field width, peak, reliability, spatial variability and the number of place fields per place cell. We performed further simulations where we varied the percentage of place cells, the occurrence of random calcium peaks and the coverage of the environment by the place fields.

**Place field width.**   The place field width was varied by varying the sigma of the Gaussian field for each place cell.

**Place field peak.**   The place field peak was varied by scaling the Gaussian model using a single scalar. This way, the shape of the Gaussian remained, while the overall peak could be increased or decreased.

**Number of place fields.**   We varied the number of place fields between 1 and 4, spaced to cover the whole environment but with no overlap. The upper limit of 4 place fields was therefore set by the size of the place field (50 cm) and the environment length (2 m).

**Reliability.**   We defined the reliability of a place cell as the probability it will have a place field in a given traversal. The probability $P_{field}$ was between 0 and 1, where $P_{field} = 0$ meant the place cells did not have a place field in any traversal, and $P_{field} = 1$ meant the place cells had

place fields in every traversal. At a probability between the 0 and 1 the place cells had place fields in a randomly selected proportion of traversals equal to $P_{field}$ times the total number of traversals.

**Spatial variability.**   We defined the variability of a place cell as the average deviation of the place field per traversal from the centre of the average place field. We modelled this by defining a Gaussian centered around the centre of the average place field, with a flatter Gaussian (i.e. higher sigma) equating to a higher variability. For each traversal we drew a value from this Gaussian distribution to be the centre of the place field for that traversal.

**Percentage of place cells.**   The percentage of cells that were designated as place cells out of 100 total cells was varied.

**Occurrence of random calcium peaks.**   We varied the occurrence of random calcium peaks in the control cells alone (S7D–S7F Fig) or all cells (S7J–S7L Fig). The occurrence in a measure of peaks per frame was measured, and this number was used as the probability for any given frame to contain a calcium peak, to create an occurrence map. This occurrence map was then convoluted with the calcium shape extracted from our real data, to create a realistic calcium peak.

**Place field coverage.**   In the default simulations the place fields of the place cells were at equidistant locations covering the entire environment. We performed an additional simulation where the place fields of all place cells were in the same location, thus not covering the entire environment.

## Performance measures

To calculate the performance of the place cell detection methods, we calculated the sensitivity and specificity of each method using the number of true positives (TP), true negatives (TN), false positives (FP) and false negatives (FN). The sensitivity is a measure of how well the method is able to find all the true place cells in the dataset, while the specificity is a measure of how specific the method is for identifying just true place cells without false positives.

$$Sensitivity = \frac{TP}{TP + FN} \tag{1}$$

$$Specificity = \frac{TN}{TN + FP} \tag{2}$$

## Place cell detection methods

We tested four methods for place cell detection (S1 Fig) all of which have all been used in previous studies to identify place cells.

**Peak method.**   This method of place cell detection was described for electrophysiology data from mice running through a 1D virtual corridor [22]. We adapted the original method for use with fluorescence data as follows: We first calculated the fluorescence maps (the average fluorescence in each location bin) for each cell from which we determined the peak fluorescence for each cell. The neuronal fluorescence data was then randomly shuffled 500 times relative to the location data by shifting the fluorescence data in time randomly by at least 5 seconds. Because the running speed of the mouse is not constant, this alters the fluorescence at each location and therefore the "peakiness" of the fluorescence map. Each cell's fluorescence map and peak fluorescence was then determined for each shuffle. Any cell with a true peak fluorescence in the top 1% of all shuffles was deemed a place cell. Thus, the peak method detects cells with high fluorescence in a place field compared to the baseline.

**Stability method.** The stability method was developed by O'Leary [23] and is based on classic methods of detecting place cells in real-world two-dimensional (2D) environments which require place cells to have a consistent place field over multiple traversals (e.g. [3]). First, a separate fluorescence map was calculated for each cell for each of the two halves of a recording session and the linear correlation between these two fluorescence maps was calculated. These correlation coefficients were then compared to control values obtained by comparing the fluorescence map of the first half of a session for that cell with the fluorescence maps of the second half of other randomly selected cells in the dataset (100 repeats per cell with redraws). The within-cell correlation of the cell was compared to all the shuffles for that cells, and if the correlation was above the 95[th] percentile of the shuffles, it was deemed a place cell.

**Combination method.** The Combination method [11,19,20] uses a combination of the level of fluorescence above baseline, with stability of firing over different traversals. It uses preprocessed fluorescence traces of cells that have been thresholded so they only include significant transients, followed by a number of thresholding steps to find "true" place fields. This method has been modified over the three publications cited above, the most recent of which was used here. First, the fluorescence map was calculated from the preprocessed traces for each cell and possible place fields were determined by thresholding the fluorescence map with a cut-off value of 0.25 times the difference between the peak fluorescence value and the baseline. Possible place fields were defined as having above-threshold fluorescence in contiguous locations for at least 20 cm and less than 120 cm. Each field was also required to have one bin with a value of at least 10% of the mean fluorescence of that cell over the session. The mean fluorescence in each place field was then divided by the mean fluorescence outside of the place field, and cells were only deemed place cells if the in-field/out-field ratio was higher than or equal to 4. Lastly, the cells were required to have a significant transient, as defined in the preprocessing thresholding step, in at least 20% of the traversals of the environment. All traces were shuffled 1000 times with respect to location and classified using the above methods. Only cells that were classified as place cells in fewer than 5% of the shuffles were considered to be true place cells. Thus, the Combination method requires place cells to significantly increase fluorescence from baseline over a contiguous place field, with some stability of responding over time.

**Information method.** This method of place cell detection has been used in freely moving mice running either through a square or circular environment [24], or a corridor [27]. The original method relied on the detection of calcium events, but to reduce the effect of such preprocessing we adapted the method to be applied directly to the relative fluorescence. As using the fluorescence directly has been shown to accurately reflect the spatial information [34], we do not expect this to negatively impact the method's ability to detect place cells. We first calculated the fluorescence maps (the average fluorescence in each location bin) for each cell from which we determined the spatial information using Eq 3, where N is the number of bins, $f_i$ is the fluorescence in bin i of the fluorescence map, and f is the average fluorescence across the map. For the spatial information we assumed a uniform occupancy (i.e. that the mouse spent the same amount of time in each location bin). The neuronal fluorescence data was then randomly shuffled 500 times relative to the location data by shifting the fluorescence data in time randomly by at least 5 seconds, the same procedure we employed for the Peak method. Each cell's fluorescence map and spatial information was then determined for each shuffle. Any cell with spatial information in the top 5% of all shuffles was deemed a place cell. Thus, the peak method detects cells with high

level of spatial information in a place field compared to the baseline.

$$SI = \sum_{i=1}^{N} f_i \log_2 \frac{f_i}{f} \qquad (3)$$

## Model cell properties

The way in which varying the different characteristics of model place cells affects their properties was illustrated (S3 Fig). By comparing real cell properties (e.g. of FWHM) to the modeled variables (e.g. width), experimenters can predict the impact of their choice of place cell classification method on detection of their real world cells.

**Full width at half maximum.** We used the full width at half maximum (FWHM) as representation of the place field width. We calculated the maximum value within the fluorescence map, and then determined the width of the place field at half this value. Notably, this value will be lower than the width defined in the model, because the FWHM is a conservative representation of the width.

**Stability.** The stability was again defined as the correlation coefficient between the fluorescence maps of the first and the second half of the session, as used in the Stability method.

**Mutual information.** Mutual information between the fluorescence traces and the location of the animal was calculated using Eq 3, as described above.

**Out/in ratio.** The out/in ratio was the mean fluorescence outside of the place field divided by the fluorescence within the place field, and is a measure of the signal to noise ratio of the place field. A lower number represents a higher signal to noise ratio.

## Place cell properties

The place cells were characterized using several of the Suite2P outputs (ROI size, average and maximum spiking rate) [31], in addition to characteristics that were calculated using custom analyses:

**Mutual information.** Mutual information between the fluorescence traces and the location of the animal was calculated using Eq 3, as described above.

**Cell-cell distance.** The cell-cell distance was calculated as the mean distance in pixels between the centre of the ROIs for all cells of one type (i.e. place cell or non-place cell) within the field of view.

**Peak intensity.** The peak intensity was the size of the largest peak (in $\Delta$F/F) in the fluorescence cell, as used in the Peak method.

**Stability.** The stability as defined as the correlation coefficient between the fluorescence maps of the first and the second half of the session, as used in the Stability method.

**Peak percentile.** The peak percentile was the percentile score of the peak intensity compared to the peak intensity of the shuffles, as used in the Peak method.

**Stability percentile.** The peak percentile was the percentile score of the stability compared to the stability of the shuffles, as used in the Stability method.

## Statistical analysis

All statistical tests were conducted in R [47]. Where appropriate, a Shapiro–Wilk test was used to test the data for normality. If the data was determined not to deviate from a normal distribution ($p > 0.05$), we performed the appropriate parametric test, otherwise a non-parametric test was applied. The tests performed for each comparison are detailed in the text. Linear mixed models were performed in R using the lme4 package (version 1.1–23). All other statistical tests were performed in R using the rstatix package (version 0.6.0).

## Supporting information

**S1 Fig. Graphical summary of tested methods.** A graphical representation of how the (A) Peak, (B) Stability, (C) Combination and (D) Information methods define a place cell based on its fluorescence map.
(EPS)

**S2 Fig. Experimental setup.** The two-photon experimental setup as seen from behind the mouse. The mouse is head-fixed and standing on a wheel that it can use to control the environment projected onto the screens in front of it.
(EPS)

**S3 Fig. Properties of model place cells.** Full width half maximum, stability, mutual information and out/in fluorescence ratio of modelled place cells with varying number of traversals (A-D), width (E-H), peak fluorescence (I-L), variability (M-P) and reliability (Q-T).
(EPS)

**S4 Fig. Model place cells of varying widths.** Fluorescence maps of model place cells with a width of (A) 20 cm, (B) 50 cm, (C) 100 cm and (D) 200 cm.
(EPS)

**S5 Fig. The effect of the number of place fields on detection.** (A) Sensitivity and (B) specificity of methods as a function of the number of place fields. Data shown are the means of 10 trials with randomly created data using the set parameters, error bars show 95% confidence interval. Sensitivity of all methods except for Combination method are 1 in A, resulting in overlapping lines.
(EPS)

**S6 Fig. The combined effect of place field width and peak.** Sensitivity (A) and specificity (B) of the Peak, Stability, Combination, and Information methods (as labelled) as a function of the peak of the place field and the width of the place field (up to 200 cm, the full environment length). Data shown are the means of 10 trials with randomly created data using the set parameters.
(EPS)

**S7 Fig. The effect of number of place cells and firing probability of non-place and place cells.** (A) Example cell population with 80% modelled place cells. Sensitivity (B) and specificity (C) of the Peak, Stability, Combination, and Information methods as a function of the percentage of modelled place cells in the population. Altering the number of non-place cells increased specificity of the Stability method (i.e. fewer non-place cells, and therefore fewer total cells, were identified) when the number of place cells was increased beyond what is likely to be physiological (20–30%, [29]). (D) Example activity of a non-place cell with 0.01 random calcium peaks per frame. Sensitivity (E) and specificity (F) of the Peak, Stability, Combination, and Information methods as a function of the activity (peaks per frame) of the non-place cells. In the originally modelled datasets, the non-place cells all contained random background levels of Poisson noise with no firing, whereas the non-place cells in real data most likely do fire, though in a location-independent manner. We therefore introduced calcium peaks in the non-place cells with shapes and numbers of peaks modelled on our real data. There was no effect on sensitivity of increasing the activity of non-place cells, but the specificity of the Stability method increased with increasing non-place cell activity. (G) Example cells population with all place fields in the same location. Sensitivity (H) and specificity (I) of the Peak, Stability, Combination, and Information methods depending on the location of the place fields, showing an

increase in specificity for the Stability method when all place cells have the same place field. This is likely due to an overall increase in correlations in the shuffled controls, as place cells will now correlate more highly with other place cells. This increase in average correlation for the shuffles, and thus an increase in threshold for being included as a place cell, increases the specificity. (J) Example activity of a non-place cell with 0.01 random calcium peaks per frame. Sensitivity (K) and specificity (L) of the Peak, Stability, Combination, and Information methods as a function of the activity (peaks per frame) of the all cells. Adding random calcium peaks to the place cells, in addition to the non-place cells, affects the specificity of the Stability method similarly to adding these peaks only to the non-place cells. This manipulation also causes an increase in sensitivity for the Combination method. These results can therefore not fully explain why the Stability method found fewer cells in our real data set. Data shown are the means of 10 trials with randomly created data. Error bars show 95% confidence intervals. Sensitivity of methods except for Combination method are 1 across the various manipulations in B, E, H and K, resulting in overlapping lines.
(EPS)

**S8 Fig. Examples of cells identified by the Peak and Information method, but not the Stability method.** Smoothed fluorescence maps of example cells that either (A) gain or (B) lose a place field, or (C) that have different place fields at different times in the session. The colour represents the normalized fluorescence.
(EPS)

**S9 Fig. The effect of altering Combination method thresholds on identification of place cells in model and real data.** (A) The sensitivity of the Combination method on detecting place cells in model data is increased when preprocessing is introduced, The sensitivity of the Combination method as a function of (B) in-field threshold (the minimum peak within a place field allowed as a fraction of the difference between the overall fluorescence peak in the data and the baseline), (C) minimum field width, (D) maximum fluorescence peak in the field, (E) in/out-field ratio, (F) ratio of traversals the cell fired in. (G) The percentage of ROIs in real data identified as place fields using Combination or Peak bootstrapping methods. (H) The percentage of ROIs identified as place cells by different methods, using the Peak bootstrapping method within the Combination method. Blue dashed lines indicate parameters used in published data and earlier analyses here. P-value in A is from an unpaired two-sample Wilcoxon test, p-value in B from a paired t-test.
(EPS)

## Acknowledgments

With thanks to Alice O'Leary and Daniel Dombeck for sharing their code. We would like to thank O. Hall-Bird and M. Hall-Bird for their insights and useful suggestions on this project.

## Author Contributions

**Conceptualization:** Dori M. Grijseels, Caswell Barry, Catherine N. Hall.

**Data curation:** Dori M. Grijseels.

**Formal analysis:** Dori M. Grijseels.

**Funding acquisition:** Catherine N. Hall.

**Investigation:** Dori M. Grijseels, Kira Shaw, Catherine N. Hall.

**Methodology:** Dori M. Grijseels, Kira Shaw, Caswell Barry, Catherine N. Hall.

**Project administration:** Catherine N. Hall.

**Resources:** Caswell Barry, Catherine N. Hall.

**Software:** Dori M. Grijseels, Kira Shaw.

**Supervision:** Caswell Barry, Catherine N. Hall.

**Validation:** Dori M. Grijseels, Caswell Barry, Catherine N. Hall.

**Visualization:** Dori M. Grijseels, Catherine N. Hall.

**Writing – original draft:** Dori M. Grijseels, Catherine N. Hall.

**Writing – review & editing:** Dori M. Grijseels, Kira Shaw, Caswell Barry, Catherine N. Hall.

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
