## [Decision Letter · Decision Letter 0]

18 Mar 2021

Dear Dr. Hall,

Thank you very much for submitting your manuscript "Choice of method of place cell classification determines the population of cells identified." for consideration at PLOS Computational Biology. As with all papers reviewed by the journal, your manuscript was reviewed by members of the editorial board and by several independent reviewers. The reviewers appreciated the attention to an important topic. Based on the reviews, we are likely to accept this manuscript for publication, providing that you modify the manuscript according to the review recommendations.

Sincerely,

Michele Migliore

Associate Editor

PLOS Computational Biology

Kim Blackwell

Deputy Editor

PLOS Computational Biology

[LINK]

Reviewer's Responses to Questions

**Comments to the Authors:**

Reviewer #1: The paper discusses methodological considerations in classifying hippocampal place-cells in two-photon calcium imaging data. The authors present a thorough analysis and discussion of 3 methods and their comparison on a modeled dataset, manipulating for several parameters that may arise in real data. Lastly, the authors compare the methods when applied on real data. They find striking differences between the methods and differing detection characteristics/overlap for place cells, highlighting the need for standardization. The article is well written and presented with only a number of minor issues affecting readability and clarity.

Comments:

In the section discussing differences between simulated vs. real data results, the authors propose that the difference in reported place-cell numbers in the stability method may be due to changing place-field characteristics over the time course of a recording. However, in simulated results from Fig 3, it was reported that stability would still report equal or higher sensitivity when variance increased or when reliability decreased. After referring to Fig. S6, the fluorescence changes that may factor into decreasing number of detected place-cells using the stability method in the real data appear to be similar to the simulated manipulations in Fig 3, yet the findings disagree. Clarification is needed.

The authors should provide further details on their linear mixed-effects modeling, including package, package version, and variables.

Fig 3, panels B, C, E, F missing legend

Fig 4, panel C. The figure indicates that this includes all cells in the recording. However, there seems to be a consistent fluorescence of location at (almost) each cell. Are these only the cells identified by the one (or all) of the detection methods? The authors should clarify.

Fig4, panel G, the image shows more areas detected as place-cells with the stability method, which doesn't agree with earlier panels. Why is that the case?

Fig S1 is quite informative but would benefit from slight clarification linking the text-boxes to exact subplots.

Fig S3, left panel is missing the peak method line. If overlapping, it should be adjusted or noted in caption.

The paper discusses differences in methodology and their critical effects on results and potential inconsistencies in replication. I agree with the authors and would strongly recommend sharing the code for the 3 compared methods. Having access to the the code will ensure that any implementation idiosyncrasies are accounted for and will be a great asset for future work and replication to supplement the model-data generation scripts they have provided.

Reviewer #2: The authors presented a robust comparison between three different methods for classifying place cells based on their calcium activities. After establishing the impact of several parameters on the performance of the three approaches in an in-silico environment, they used them to classify place cells from the CA1 distal portion of the hippocampus of mice stimulated in a 1D VR environment. They found that the three methods identified nearly independent population of place cells in vivo, raising concerns about what we should use to classify and how we should define place cells. Overall, the manuscript is linear, clear and relevant. I have only minor comments and clarifications.

• Could you please provide the codes of the different methods? In alternative, pseudocodes, in addition to Suppl. Fig.1 (which is very clear), could help the reader to easily reproduce your approaches.

• It is not clear to me how much of the “excess activity variance” behavior (place cells also fire outside the place field) was modeled in the in-silico dataset.

• Considering that a place cell fires about 50% of the time spent in the field place, how this variability was introduced in the in-silico dataset? If it was not included explicitly in the model, do you think that it could affect the performance of the Peak method in particular?

• You modeled the place cells as independent from each other and from the non-place cells. Considering that the network state is predictive of the place field response of a place cell, how much do you think that the performances of each method were biased by the absence of a network?

• In consideration of the role of theta rhythms on the activity of the place cells, in which measure do you think that the presence of such signature could affect the performance of the methods?

• Considering the two previous comments, could you comment about your decision to omit network behaviors in the model? I understand that the simulations must be simple and network effects are difficult to model, yet it would be important underline this lack in the manuscript in my opinion.

• It is unclear to me if, in the Peak method, the rate map across locations is also calculated across transversal; since in Fig.3B the sensitivity of the Peak method showed a dependence to the % of reliability, I assume that the transversal are somehow considered. Could you please clarify it?

• It is not clear to me how in the Stability method non-place cells can be detected (specificity about 80%) in the in-silico models if they are not firing. Do you have any explanation or am I missing something?

• In the in-silico dataset, the authors modeled 100 cells, which is reasonable and in line with the average number of neurons detected in vivo using optical approaches. In which measure the Stability method is affected by the number of total cells (since random cells are selected in the method)? Could this partially explain some decays in the performance of the approach? Could you comment?

• For a given simulation, was each parameter the same across the place cells or it ranged (i.e. width=50 ± xx cm)? In other words, has the same Gaussian field been used for all 20 cells for a given simulation? If so, how much do you think that this homogeneity could bias the performances of the methods?

• Could you clarify in the manuscript the reasons why you used the 1D Gaussian model of the fluorescence of a place field? How much do you think that the use of a Gaussian bias the Peak method in particular?

• Is the activity of the nth transversal modeled independently from the history of the activity of the nth-1 transversal? Do you think that activity-dependent effects could impact the performance of each method?

• The authors distributed the place field centers evenly across the map; do you think that this could have an effect on the performance of the Stability method since each cell would have a significant overlap (higher correlation than random non-place cells or farther place cell in the field) of their place fields with their neighbors (Step 3 of the method)?

• Was the noise added in the model able to influence the Gaussian field distribution and, more broadly, the probability of a cell to fire or not?

• Did you perform a set of simulations in which the Gaussian was different for each cell, thus varying in a narrow range width, peak, % reliability and % variability? Even though varying one parameter at time is instrumental in dissecting single contributions, do you think that it could be beneficial to model a more realistic scenario to test the performance of the different methods? You did already something very similar in Supplementary Figure 4 but varying only width and peak. It would be interesting to understand in this scenario the percentage of cells identified from each method and the percentage of cells identified from more than one method (and maybe help to clarify also the discrepancies highlighted in the real data).

• In the results section the authors correctly pointed out that the spatial information carried by place cells detected with the Peak method can derive also from other sources, yet in the discussion the repeatedly used this argument in favor of the Peak method. It sounds odd to me (and I personally agree with the caution mentioned in the results section).

• In the in vivo experiments each mouse had 2 sessions; did a session-dependent effect emerge in the place cell classification?

• Personal note: for a given figure, in my opinion it would be clearer to have both the sensitivity and specificity graph y-axis ranging from 0 to 1. It could give the reader an immediate sense of the fact that usually specificity is more robust.

Reviewer #3: Summary:

Place cells are hippocampal pyramidal neurons whose activity is modulated by the position of an animal in its environment. Collectively, an ensemble of place cells can be used to decode the position of an animal in its environment. Recently new calcium imaging techniques have been used to study place cells. They notably allow direct cell visualization, facilitating detection across several days or even weeks and recording of a large neuronal population (up to a thousand cell). However, calcium indicators only indirectly report single cell activity and this could dampen proper analysis of place cells.

Several methods can be used to detect place fields ranging from simple ones based on a fixed threshold applied to activity averaged over several trials to more sophisticated ones based on comparisons with shuffled data. The present study aims at comparing different methods on a model dataset and real calcium imaging data. We believe this approach could be useful to decide which method should be used depending on the specific experimental design.

1. Authors should justify the three methods compared. They do not appear to be widely used in the field. One (The combination method) was used essentially by a single lab (Tank lab) another one (the stability one) was used only in an unpublished work (O’Leary et al., SFN abstract 2019) and the third one (the peak method) has been used only in one study. To be useful for the community I think this work should include more widely used methods for detecting place fields in calcium imaging data (e.g. from the Ziv or Lonsonczy labs).

2. As discussed in the manuscript the ultimate question is what is a place cell (what are their core properties) with detection methods differing accordingly. The most prevalent property of place cells is that their activity should be modulated by the position of the animal in the environment in a consistent way. So, a bump of activity should clearly be observed specifically and consistently when animal is in a specific location in the track. In terms of consistency, the combination method whose sensitivity slowly builds up with increasing consistency looks more appropriate. Could the authors comment on that?

3. Along the same line place cells with a place field width as large as the all environment (Fig. 2) probably convey few information about animal location and it is questionable whether these cells can be called place cells. In that respect a method like the combination method that does not detect place cells with very large place fields is probably more relevant than methods that readily detect those place cells but this interpretation is not conveyed in the manuscript nor discussed.

4. The drop of specificity of the stability method with the number of traversals is puzzling. From the example cells in Fig. 1E firing appears to be very stable between two halves of the recording for at least 50 trials. Why is this drop in specificity observed?

5. It would be useful to have more quantitative data on the synthetic place cells used in Figs 1 and 2 beside the size of the place field for example of the amount of out versus in field firing, the reliability and spatial variability. It should ideally be comparable with place cells reported with electrophysiological recordings in rats and mice.

6. The authors state that detection methods used with electrophysiological recordings cannot be adapted to data obtain with imaging techniques but it would be interesting for readers to determine how detection methods more classically used with electrophysiological recordings perform on the model dataset and how the population of detected place cells overlap with that of place cells detected with other methods. It would also widen the impact of the paper by extending its conclusions to data obtained with electrophysiological recordings.

7. Some imaging techniques use deconvolution to infer spikes trains from imaging data. It is unclear how good are spike train inferred from imaging and deconvolution methods but recent report suggests that they could be quite accurate. Then an important question is whether place cell detection would be better from the deconvoluted spike trains than from raw calcium data. Could the authors use their model dataset to provide answers to this question?

8. Often a fixed velocity threshold is used to separate periods of movements (included in place cells analysis) from periods of immobility (excluded from place field analysis). Notably some of the methods analyzed here are performed using a fixed threshold. It is thus unclear why a variable threshold (10 % of maximum velocity) was used. Depending on the running behavior of the mice this will lead to highly different thresholds for movement detection (for example 4 cm/s in a fast running mice/session compared to 0.5 cm/s in a slow running mice/session). What is the impact of fixed versus variable velocity threshold on the results.

Minor:

1. Fig. 1 could you illustrate trial by trial activity of non-place cells for comparison?

2. Fig. 2 could you show example of place fields with different width including 200 cm width.

**Have all data underlying the figures and results presented in the manuscript been provided?**

Reviewer #1: Yes

Reviewer #2: Yes

Reviewer #3: Yes

PLOS authors have the option to publish the peer review history of their article (what does this mean?). If published, this will include your full peer review and any attached files.

Reviewer #1: No

Reviewer #2: **Yes: **Mattia Bonzanni

Reviewer #3: **Yes: **Jérôme Epsztein

Figure Files:

Data Requirements:

Reproducibility:

References:

---

## [Editor Report · Decision Letter 1]

15 Jun 2021

Dear Dr. Hall,

We are pleased to inform you that your manuscript 'Choice of method of place cell classification determines the population of cells identified.' has been provisionally accepted for publication in PLOS Computational Biology.

Best regards,

Michele Migliore

Associate Editor

PLOS Computational Biology

Kim Blackwell

Deputy Editor

PLOS Computational Biology

---

## [Editor Report · Acceptance letter]

2 Jul 2021

PCOMPBIOL-D-21-00323R1 

Choice of method of place cell classification determines the population of cells identified.

Dear Dr Hall,

I am pleased to inform you that your manuscript has been formally accepted for publication in PLOS Computational Biology. Your manuscript is now with our production department and you will be notified of the publication date in due course.

With kind regards,

Katalin Szabo
